# Specifying the light absorbing properties of aerosol particles in fresh snow samples, collected at the Environmental Research Station Schneefernerhaus (UFS), Zugspitze

Martin Schnaiter[1,4], Claudia Linke[1], Inas Ibrahim[1], Alexei Kiselev[1], Fritz Waitz[1], Thomas Leisner[1], Stefan Norra[2], and Till Rehm[3]

[1]Institute of Meteorology and Climate Research, Atmospheric Aerosol Research, KIT, Karlsruhe, Germany
[2]Institute of Geography and Geoecology, KIT, Karlsruhe, Germany
[3]Environmental Research Station Schneefernerhaus (UFS), Zugspitze, Germany
[4]schnaiTEC GmbH, Karlsruhe, Germany

*Correspondence to*: Martin Schnaiter (martin.schnaiter@kit.edu) and Claudia Linke (Claudia.Linke@kit.edu)

## Abstract

Light absorbing impurities in snow result in a darkening of the Earth's white snow and ice covers. Airborne particles like mineral dust, volcanic ash or combustion aerosol particles are able to reduce the snow and ice albedo considerably with a very small mass fraction of deposited particles. In this study, a new laboratory method is applied that allows to measure the spectral light absorption coefficient of airborne particles that are released from fresh snow samples by an efficient nebulizing system. Three-wavelength photoacoustic absorption spectroscopy is combined with refractory black carbon (BC) mass analysis to determine the snow mass specific and the BC mass specific absorption cross sections. Fullerene soot in water suspensions are used for the characterization of the method and for the determination of the mass specific absorption cross section of this BC reference material. The analysis of 31 samples collected after fresh snowfall events at a high-altitude alpine station reveal a significant discrepancy between the measured snow mass specific absorption cross section and the cross section that is expected from the BC mass data, indicating that non-BC light absorbing particles are present in the snow. Mineral dust and brown carbon (BrC) are identified as possible candidates for the non-BC particle mass based on the wavelength dependence of the measured absorption. For one sample, this result is confirmed by environmental scanning electron microscopy and by single particle fluorescence measurements, which both indicate a high fraction of biogenic and organic particle mass in the sample.

## 1. Introduction

Light absorbing atmospheric particles like black carbon (BC), brown carbon (BrC), mineral dust or volcanic ash are eventually removed from the atmosphere by dry and wet deposition. Light absorbing particles that are deposited into snowpack result in a darkening of the otherwise white surface by increased absorption of solar radiation. Because pure snow is the most reflective natural surface on earth, the presence of small amounts of absorptive impurities changes the optical properties of snow resulting in a considerable reduction of the snow albedo (Warren, 1982; Warren and Wiscombe, 1980). In cold fine-grained snow observed BC amounts of $\leq 100$ ng/g can reduce the spectral albedo at visible wavelengths by up to 2%, while in melting snow, these reductions can even increase to 6% (Warren and Wiscombe, 1985). Although these numbers seem to be small, they result in a significant impact on the radiation budget via the snow albedo effect (Clarke and Noone, 1985; Flanner et al., 2007), which also

includes secondary effects like rapid snow transformation (e.g. changes in the snow grain size) and the retreat of snow and ice covers in a warming climate. The IPCC reported a global annual mean radiative forcing for anthropogenic BC in snow and ice of + 0.04 W m$^{-2}$ with an uncertainty range of 0.02 to 0.09 Wm$^{-2}$ (Boucher et al., 2013). A considerably higher radiative forcing estimate for BC in snow is given in Bond et al. (2013). In their estimate they calculated an effective BC in snow forcing that includes feedback mechanisms, like rapid adjustments of the snowpack as well as the climate response to the snow and ice albedo changes, resulting in a best forcing estimate of + 0.13 W m$^{-2}$ with an uncertainty of 0.04 to 0.33 W m$^{-2}$.

For a reliable assessment of the radiative forcing by light absorbing impurities in snow and ice, the response of the snow albedo to the presence of light absorbing particles has to be understood from a physical basis. This is not a trivial task as the spectral albedo of snow depends not only on the mass mixing ratio of the absorbing particles in the snowpack, but also on the chemical composition, the microphysical properties, as well as the spectral absorption properties of the particles in addition to the snow grain size distribution and the spatial variations of these parameters (Flanner et al., 2007). While the mass mixing ration of BC in snow (and also that of mineral dust and organic carbon) has been the subject of many recent studies (e.g. Kaspari et al., 2014; Schmale et al., 2017; Zhang et al., 2018), their microphysical as well as spectral optical properties are still poorly investigated. A few studies exist on the microphysical (Dong et al., 2018; Zhang et al., 2017) as well as optical properties (Dal Farra et al., 2018; Doherty et al., 2010; Kaspari et al., 2014; Schwarz et al., 2013), but these are too sparse from giving a conclusive picture. Further, the optical properties of light absorbing impurities are often indirectly addressed by either applying simplified optical particle models (e.g. Schwarz et al., 2013) or by using mass specific absorption cross sections (MAC) determined for atmospheric particles (e.g. Dong et al., 2018; Zhou et al., 2017). There is a need for more studies that address the question on the microphysical nature and the optical properties of particles in snow and ice. In a large-area study on light absorbing impurities in Arctic snow Doherty et al. (2010) applied the integrating-sandwich with integrating sphere technique (ISSW; Grenfell et al., 2011) to measure the snow mass specific spectral absorption cross section, $\sigma_{abs}$, on filter samples. The ISSW is an improved version of the original integrating plate (IP) filter method that was used in the pioneering work of Clarke and Noone (1985) to determine the BC in the Arctic snowpack. Although their method measures $\sigma_{abs}$, they used a fixed MAC for BC and assumed fixed absorption Angström exponents for BC and non-BC to calculate the BC mass mixing ratios and the fraction of non-BC particles in the snow, respectively. However, large biases have been revealed in laboratory tests of the ISSW when non-BC absorbing and purely light scattering particles are co-deposited on the filter (Schwarz et al., 2012). This indicates a significant cross-sensitivity of the ISSW-determined $\sigma_{abs}$ to particle light scattering.

In the present study we used a different approach to measure $\sigma_{abs}$ by applying three-wavelength photoacoustic spectroscopy on re-aerosolized snow samples. Photoacoustic aerosol spectroscopy is not prone to light scattering artefacts and, therefore, gives reliable results also in the presence of light scattering impurities. As a pilot study for this approach, freshly fallen snow was collected at the alpine research station Schneefernerhaus over a period of six months in Winter 2016/2017. Particulate impurities in the samples were re-aerosolized in the laboratory and were concurrently measured by a single particle soot photometer (SP2) as well as by a homebuilt three-wavelength photoacoustic spectrometer (PAAS-3λ) to determine $\sigma_{abs}$ and the BC mass specific MAC of the snow impurities. Suspensions of Fullerene soot in water were prepared as a standard for melted snow and were used to characterize the nebulizer efficiency and to determine the PAAS-3λ detection sensitivity. From these measurements, also the MAC of the Fullerene soot standard was deduced and was compared with the MAC of the snow samples.

Sample preparation, test particle properties and experimental setup are given in Sect. 2. Sect. 3 describes the characterization of the nebulizer at different operational settings and gives the efficiency of the nebulizer in terms of particle number and mass concentrations. The MAC of the Fullerene soot standard is presented in Sect. 4. Measurement results of the snow samples are discussed Sect. 5 followed by conclusions given in Sect. 6.

## 2. Experimental

### 2.1 Snow samples

The snow samples used in this study originate from the Environmental Research Station Schneefernerhaus (UFS). The station is located at a latitude of 47° 25' 00'' N, a longitude of 10° 58' 46'' E and an altitude of 2650 m a.s.l. During winter the UFS is situated in the free troposphere most of the time with low boundary layer influence (Gilge et al., 2010) but can be occasionally affected by long range transported Saharan dust plumes (Flentje et al., 2015) or in rare events by transatlantic aerosol transport (Birmili et al., 2010). However, the station is located within a skiing and hiking resort and is therefore sporadically affected by local anthropogenic emissions during daytime, e.g. from snow groomers during the skiing season. Yuan et al. (2019) report multiple short-term atmospheric CO events and higher atmospheric NO peaks during the weekdays (mostly around 09:00 LT) at the station. Disregarding these local pollution events at the station, the multi-annual median BC mass concentration is 0.1 μg/m³ and is typically well below this value during the winter months (Sun et al., 2019).

Snow samples were collected during a time period from December 2016 to May 2017. During this time period basic meteorological data of precipitation, maximum and minimum temperatures, sunshine duration and wind speed are provided by the German Meteorological Service (DWD). Aerosol parameters like number concentrations from condensation particle counter (CPC) and optical particle counter (OPC) instruments, equivalent BC mass concentrations from multi-angle absorption photometer (MAAP) measurements as well as dust loads are provided by the German Federal Environment Agency (UBA) and the German Meteorological Service (DWD). The snow samples were taken either during or just after snowfall events by scraping off only the top few centimeters of the snowpack to avoid sampling older snow. A metallic hand shovel is used to sample the snow from an area of about 30 x 30 cm into a zipper sealed polyethylene household plastic bag with a volume of 1 L (Toppits, Germany). In this way, snow from the beginning of the snowfall event could be missed, but most of the time the events were accompanied by heavy wind, so that it was impossible to completely sample the fresh snow layer. After collection, the samples were stored at the UFS in a freezer at -18°C until they were transported under frozen conditions to the laboratory at KIT. During six months, 33 samples were taken at the UFS.

Right before the analysis, which is described in the following sections, approximately 30 mL snow is detached from the plastic bag and is put in a glass beaker for further treatment. This subsample is then melted by sonication in an ultrasonic bath (EMAG Technologies, Germany) at room temperature for about 5 minutes. Sonication during the melting process should help to avoid particle adhesion to the wall of the glass beaker. Aqueous snow/ice sample sonication prior to the analysis is recommended by several groups (e.g. Kaspari et al., 2011; Wendl et al., 2014) although with inconclusive results of the obtained improvements. The melted samples were never refrozen for later analysis as this can result in a significant particle mass loss of up to 60% (Wendl et al., 2014).

### 2.2 Setup of instruments

The experimental setup consisting of a Marin-5 Enhanced Nebulizer System (Teledyne CETAC Technologies, USA), a self-built three-wavelength photoacoustic aerosol absorption spectrometer (PAAS-3λ, Linke et al., 2016) and a single particle soot photometer (SP2, Droplet Measurement Technologies, USA) is shown in Figure 1. The liquid samples are fed by a peristaltic pump (ISM795, ISMATEC, Wertheim, Germany) equipped with Tygon tubing (ID 0.76 mm, E-LFL, Fisher Scientific, USA) into a concentric pneumatic glass nebulizer ("MicroMist 500" with a critical orifice of 500 μm), which is positioned inside of the Marin-5 nebulizer. The glass nebulizer consists of a capillary that directs the liquid sample to the nebulizer tip where a concentric sheath flow of pressurized synthetic air disperses the supplied liquid sample into a spray (Katich et al. 2017). The optimum specified liquid sample flow rate of the glass nebulizer, which is set on the peristaltic pump, is $R_{pp}$=0.5 mL/min. The generated spray is released into the nebulizing chamber of the Marin-5 nebulizer with the maximum specified gas flow rate of $R_{neb}$=1.1 L/min (see Figure 2). The spray enters the heated section of the chamber where the water is evaporated forming a moist aerosol of residual particles. This aerosol comprises refractory aerosol particles and residues from soluble material that were contained in the liquid sample. Subsequently, the aerosol passes the cooled section of the chamber where excess water is eliminated from the sample flow by condensation to the chamber walls, so that the bulk of water vapour in the moist aerosol is drained before the particles leave the nebulizer. The aerosol that exits the nebulizer is then directed through a homebuilt silica gel dryer to reduce the relative humidity of the aerosol flow below 20% r.h. Finally, sample flows for the SP2 and the PAAS-3λ are taken from the aerosol flow, which is otherwise exhausted to the ambient.

The SP2 was used to determine the refractory black carbon (BC) mass concentration $c_{BC}^{SP2}$ of the BC suspension standards and the snow samples. The instrument is typically operated with a sample flow rate of 0.12 L/min. For single particle mass determination, the two incandescence channels of the instrument are calibrated with Fullerene soot particles (Alfa Aesar, #40971, lot #F12S011) size selected by a differential mobility analyser (DMA) in the range of 100 nm to 450 nm, corresponding to single particle refractory BC masses of about 0.5 fg to 30 fg. Note that this "Fullerene soot" material actually contains only a minor mass fraction of Fullerenes of less than 10% (Laborde et al., 2012a). The gains of the two incandescence channels of the SP2 are adjusted in a way that they cover a combined volume-equivalent size range from 60 to 560 nm (Laborde et al., 2012b). For quantifying the number efficiency of the nebulizer, number concentrations of non-absorbing PSL particles are determined from the SP2 single particle scattering data. For analysis of the SP2 data, the software toolkit developed and provided by Martin Gysel from the Paul Scherrer Institute, Switzerland is used (Gysel et al., 2011).

The PAAS-3λ is a single cavity three-wavelength photoacoustic aerosol absorption spectrometer, which has been designed and built at KIT and which is currently marketed by schnaiTEC GmbH. The instrument concept and working principle are described in detail in Linke et al. (2016). Briefly, to measure the absorption coefficient $b_{abs}$ of aerosols, a controlled sample flow of 0.85 L/min is led through the acoustic resonator of the instrument. This open-ended cylindrical cavity has a diameter of 6.5 mm and a length of 49 mm resulting in a fundamental acoustic resonance frequency of about 3200 Hz. Acoustic buffers of 24.5 mm length and 78 mm diameter are attached to both ends of the cavity to filter acoustic disturbances that may exist in the frequency range of the resonator. Possible disturbances comprise noise generated in the flow system as well as ambient sound. The photoacoustic cell composed of the acoustic resonator and the buffers has a total volume of 236 cm³. The PAAS-3λ system was developed to determine the absorption coefficients at three wavelengths across the visible spectral range. Three lasers (Dragonlaser, Changchun, China), modulated at the resonance frequency of the acoustic cavity, are used in

this study. The lasers have emitting wavelengths of 405, 532, and 658 nm and generate modulated emission power of 100, 150 and 130 mW, respectively. The modulation frequency has a duty cycle of 50% and is tuned to the resonance frequency of the acoustic cavity on a daily basis. The detection limit ($2\sigma$) of this setup was derived from an Allan deviation analysis of long-term background signal measurements similar to the analysis presented in Fischer and Smith (2018). The Allan deviation plot for the three PAAS-3λ wavelengths is shown in Figure 3. According to this analysis, the PAAS-3λ instrument has a $2\sigma$ detection limit in the range of 1.2 to 2.1 1/Mm for all three wavelengths and for a typical averaging time of 60 seconds.

### 2.3 PSL particles

The characterisation of the particle number efficiency of the nebulizer and the daily performance control was performed with monodisperse polystyrene latex (PSL) particles (Postnova Analytics GmbH, Landsberg am Lech, Germany) with nominal diameters of $240 \pm 5$ nm and $304 \pm 5$ nm. The particle number concentration within these suspensions is about $3 \cdot 10^8$ 1/mL. A diluted PSL standard suspension sample was prepared daily by pipetting 1 mL suspension into a 100 mL graduated flask filled with Nanopure water.

### 2.4 Fullerene **Soot Standards**

Suspensions of known Fullerene soot mass concentrations were prepared to determine the particle mass efficiency $\varepsilon_{neb}$ of the nebulizer. The material has been widely used to calibrate SP2 instruments (including the present study) as it gives a sensitivity that is similar to Diesel soot (Laborde et al., 2012a). A stock suspension of Fullerene soot particles (Alfa Aesar, USA; stock #40971, lot #F12S011) suspended in Nanopure water was stored in a 250 mL graduated glass bottle with polypropylene cap (Simax, Czech Republic) for several days to allow larger particles to settle out of the suspension. From this stock suspension the supernatant suspension was taken to (a) determine the Fullerene soot mass concentration gravimetrically and to (b) prepare diluted Fullerene soot standard suspensions for daily particle mass efficiency control of the nebulizer.

For the gravimetric analysis of the settled stock suspension two empty quartz fibre filters (MK360, Ahlstrom Munksjö, Finland) were dried over night at 50°C, stored in a dehydrator for 2 hours and weighed with a microbalance (M3P, Sartorius, Germany). Then 30 mL of the supernatant suspension were extracted from the stock Fullerene soot suspension and were dropped on both quartz filters. The filters were then temperature treated the same way as the empty filters before being weighed again. From the gravimetric Fullerene soot mass, the mass concentration of the stock suspension was determined to be $5.8 \pm 2.6$ µg/mL (mean $\pm 1\sigma$; N=2). Four Fullerene soot standard suspensions for the determination of the mass efficiency $\varepsilon_{neb}$ of the nebulizer were prepared on each measurement day from the stock suspension. This was performed in two dilution steps resulting in samples of Fullerene soot suspended in Nanopure water with nominal mass concentrations of $c_{FS}= 11.5 \pm 2.5, 23 \pm 7.3, 34.5 \pm 6.3$ and $46 \pm 7.7$ ng/mL. Note that the given uncertainty range is based on the analysis of the SP2 mass measurements acquired during the characterization of the Marin-5 nebulizing efficiency (Sec. 3) and does not reflect possible systematic biases in the above gravimetric analysis of the stock suspension and the subsequent dilution process.

### 3. Characterization of the nebulizer

Particles suspended in liquid samples are not completely dispersed during the nebulizing process in the Marin-5 nebulizer but are partially lost in the drain water of the instrument. In order to characterize the Marin-5 dispersion efficiency the nebulizer settings were varied while measuring the particle output by a CPC. In this characterization the operation recommendations given by Katich et al. (2017) for applying the Marin-5 in snow sample analyses were mainly adopted. However, a minimum gas flow rate of $R_{neb}$=0.97 L/min is necessary in the present study to simultaneously operate the SP2 and the PAAS-3λ downstream the Marin-5. Also, the relative humidity of the Marin-5 output flow is an important property here as reliable photoacoustic measurements of aerosol systems require a relative humidity below ~30% (Langridge et al., 2013). The particle output (in terms of particle number and particle mass) and the relative humidity depend on the $R_{pp}$ and $R_{neb}$ flow rates and the temperatures of the heated and cooled sections of the nebulizing chamber. As a starting point of the Marin-5 characterization the nebulizer parameters were set to heating and cooling temperatures of 120°C and 5°C, respectively, the maximum specified input air flow rate of $R_{neb}$=1.1 L/min, and a liquid sample flow rate of $R_{pp}$=0.08 mL/min. Each parameter, except $R_{neb}$, was varied while keeping the others constant. It turned out that the temperature of the cooled section of the chamber has only a minor influence of a few percent on the dispersion efficiency and the relative humidity in the output flow.

In Figure 4, the PSL particle number nebulizing efficiency and the relative humidity of the Marin-5 exit flow are shown as a function of the liquid sample flow rate $R_{pp}$ of the nebulizer. From Figure 4 it is clear that the higher the applied sample flow rate $R_{pp}$ the higher the relative humidity at the exit at a fairly constant efficiency of around 36%. Note that the observed constant nebulizing efficiency reflects a nearly linear correlation between the PSL number concentration at the nebulizer exit and the liquid supply rate $R_{pp}$ at the input. Even for the lowest $R_{pp}$ of 0.08 mL/min the relative humidity approaches 50% and is therefore far above the threshold humidity of 30% for unbiased photoacoustic measurements. Therefore, the silica gel aerosol dryer downstream the Marin-5 is a prerequisite in the Fullerene soot and snow sample analysis with the PAAS-3λ instrument. The PSL number concentration and relative humidity behave in a similar way when $R_{pp}$ is kept constant at 0.16 mL/min and the temperature of the heated section of the chamber is varied between 110 and 150°C, i.e. higher values with higher temperatures (not shown here). As a result of this characterization the liquid sample flow rate was set to $R_{pp}$=0.32 mL/min in the Fullerene soot and snow sample analyses in order to generate a sufficient absorption signal in the PAAS-3λ while the aerosol can still be dried to a relative humidity below 30%. The nebulizer temperatures were set to 120°C and 5°C for the heated and cooled section, respectively.

With the above settings and the setup shown in Figure 1, the mass nebulizing efficiency $\varepsilon_{neb} = c_{BC}^{SP2}/c_{FS} \cdot R_{neb}/R_{pp}$ of the Marin-5 nebulizer was derived from measurements using the Fullerene soot standard suspensions described in Sec. 2.4. In Figure 5 the detected SP2 mass concentrations $c_{BC}^{SP2}$ are shown for the four Fullerene soot standards which were daily prepared. The mass nebulizing efficiency was determined to be 39% from these measurements, which is in a very good agreement with the findings of Katich et al. (2017) for similar settings.

Figure 6 shows the average mass size distributions of the four Fullerene soot suspension standards used in the characterization of the Marin-5 mass nebulizing efficiency $\varepsilon_{neb}$ shown in Figure 5. Each size distribution is an average over 8 individual suspensions that were prepared on a daily basis. The measured mass size distribution of

each of the 32 individual suspensions was fitted by a lognormal function to get the mass-equivalent median
diameter (MMD), the width of the distribution (i.e. geometric standard deviation $\sigma_g$), as well as the integrated
mass concentration, $c_{BC}^{SP2}$. Note that the integrated mass concentration $c_{BC}^{SP2}$ from the lognormal fit was used in
determination of $\varepsilon_{neb}$ and the Fullerene soot mass absorption cross sections, $MAC_{FS}$, in Sec. 4. This is necessary
as the summed particle mass from the SP2 measurement alone ignores particles with sizes larger than 560 nm,
which represent a mass fraction of about 10% (see Figure 6). The suspensions show a very stable MMD of 227 ±
3.7, 226 ± 1.7, 228.5 ± 2.5, and 229 ± 1.7 nm and $\sigma_g$ of 1.57 ± 0.018, 1.56 ± 0.024, 1.57 ± 0.017, and 1.57 ± 0.016
for the 11.5, 23, 34.5, and 46 ng/mL suspensions, respectively with a MMD of 228 ± 3 nm and a $\sigma_g$ of 1.57 ±
0.018 when averaging over all 32 samples (Table 1).

## 4. Specific mass absorption cross sections (MAC_FS) of Fullerene soot

Simultaneously to the BC mass concentration measurements with the SP2, the absorption coefficients $b_{abs}$ of the
Fullerene soot suspensions were measured for the three PAAS-3λ wavelengths. Both measurements together
enable the determination of the mass specific absorption cross section $MAC_{FS} = b_{abs}/c_{BC}^{SP2}$ of airborne Fullerene
soot at 405, 532, and 658 nm. In Figure 7, the absorption coefficients $b_{abs}$ are plotted against the SP2-derived BC
mass concentrations $c_{BC}^{SP2}$ of the Fullerene soot suspension standards. Linear regression fits of the data result in
$MAC_{FS}$ values of 10.5 ± 3.2 m²/g, 9.5 ± 2.2 m²/g, and 8.6 ± 3.3 m²/g for 405, 532, and 658 nm, respectively. The
$MAC_{FS}$ at 532 nm is comparable to the value of 8.84 m²/g given by Schwarz et al. (2012) for Fullerene soot (lot
#F12S011) deduced from ISSW measurements, but is significantly higher than the 6.1 ± 0.4 m²/g (mean ± 2σ)
measured recently by photoacoustic absorption spectroscopy for size selected Fullerene soot particles by
Zangmeister et al. (2018). They used a combination of a differential mobility analyzer (DMA) and an aerosol
particle mass analyzer (APM) to select Fullerene soot particles within a narrow mass range from aerosol generated
by an atomizer. Their $MAC_{FS}$ of 6.1 m²/g, which is given for a wavelength of 550 nm, a mobility-equivalent
diameter of 350 nm, and for a particle mass of $16.6 \cdot 10^{-15}$ g, corresponds to a volume-equivalent diameter of 264
nm using a density of 1.72 g/cm³ of Fullerene soot (Kondo et al., 2011). Although, this diameter is not very
different to the MMD of 228 nm of the Fullerene soot suspensions used here, part of the observed discrepancy can
be attributed to the different sizes as the MAC is strongly depending on the particle diameter for particles larger
than about 200 nm (e.g. Moosmüller et al., 2009). To be comparable, we measured the $MAC_{FS}$ of size selected
Fullerene soot particles in a separate study by adding a DMA behind the Marin-5 in the setup shown in Figure 1.
A $MAC_{FS}$ of 8.6 m²/g was measured for the mobility-equivalent diameter of 350 nm, which is still ~40% larger
than the $MAC_{FS}$ given by Zangmeister et al. (2018) for the same diameter. However, they used an APM to measure
the BC mass, while a SP2 was used here to deduce the refractory BC mass. According to Laborde et al. (2012a),
the Fullerene soot product shows a variability from batch to batch, which results in a SP2 calibration uncertainty
of up to 15% (actually only two batches were compared; lot #F12S011 and lot #L18U002). They explained the
differences in the SP2 response (i.e. the calibration curves) by a substantial non-refractory coating in case of the
L18U002 batch that could be identified by thermodenuding the samples. Assuming that lot #W08A039 used in
Zangmeister et al. (2018) has a similar coating, this would increase the APM mass measurement by about 15%
compared to the SP2-derived BC mass of lot # F12S011 used in the present study. This in turn would increase the
$MAC_{FS}$ from 6.1 m²/g reported by Zangmeister et al. (2018) to about 7 m²/g when using only the refractory BC

mass fraction in the calculation of the $MAC_{FS}$. This assumption reduces the discrepancy between the two $MAC_{FS}$ values to 35%, which is within the uncertainty range of $\pm 2.2$ m²/g for our 532 nm value. It is further conceivable that different batches of the Fullerene soot material have different electronic band structure (i.e. refractive index) and/or fractal aggregate structures that both change the absorption cross section of the particles at a constant particle mass (e.g. Liu et al., 2019; Zangmeister et al., 2018). Figure 17 shows an electron micrograph of a typical Fullerene soot aggregate sampled from the dry aerosol output of the Marin-5 nebulizer. Thus, the Fullerene soot particles do not have a simple fractal aggregate structure, but are rather complex-structured with polydisperse monomer sizes, monomer nonsphericity (irregularity), necking, and overlapping, which all have a significant impact on the optical particle properties (including the absorption cross section) compared to the idealized fractal aggregate (Teng et al., 2019). Since these microphysical details of the soot particles are very sensitive to the actual formation and subsequent treatment conditions (Gorelik et al., 2002), it is conclusive that the $MAC_{FS}$ has an even higher variability between different Fullerene soot batches compared to what is expected from the SP2 mass sensitivity only.

The wavelength dependence of the aerosol light absorption, expressed by the so-called absorption Angström exponent (AAE), was determined to be $0.46 \pm 0.07$ for the used Fullerene soot suspensions by analysing the $b_{abs}$ data for the 405 and 658 nm wavelengths. This AAE is close to the ~0.6 reported by Baumgardner et al. (2012) for Fullerene soot derived from multiwavelength PSAP and Aethalometer measurements and it is within the range of the $0.54 \pm 0.06$ determined by Zhou et al. (2017) from ISSW spectrometer measurements on Fullerene soot filter samples in the 450 nm to 750 nm spectral range. However, it is significantly lower than the $0.92 \pm 0.05$ given by Zangmeister et al. (2018) for Fullerene soot lot #W08A039. Here again, we have to take into account that Zangmeister et al. (2018) analyzed size segregated absorption spectra and their AAE is given for a mobility-equivalent diameter of 350 nm. Analyzing our size segregated measurements gives an AAE of $0.82 \pm 0.02$ for the same mobility-equivalent diameter, which is close but smaller than the Zangmeister et al. value, further supporting the above assumption that there is a difference in the chemical as well as physical (including optical) properties between different batches of the Fullerene soot product.

In conjunction with the $b_{abs}$ detection limit of 2.1 1/Mm given in Sec. 2.2 for the PAAS-3λ and the mass nebulizing efficiency $\varepsilon_{neb}$ of the Marin-5 nebulizer given in Sec. 3, the $MAC_{FS}$ analysis shown in Figure **7** can be used to assess the detection limit of the PAAS-3λ in terms of the BC mass mixing ratio in the snow. A conservative estimate that also accounts for the uncertainties in the preparation and quantification of the Fullerene soot suspension standards gives a lower BC mass mixing ratio of 4 ng/mL that can be optically detected by the PAAS-3λ within the setup shown in Figure **1**. Therefore, the method presented here should be suitable to analyse the visible light absorption of BC snow impurities for continental as well as for the most of the Arctic areas (e.g. Table 1 in Warren (2019)).

## 5.  Results and discussion of the snow sample measurements

The instrumental setup was used to measure a set of 33 snow samples from the UFS in the same way as the Fullerene soot standards before. The results of two samples were discarded from data presented here, because they show inexplicable high BC mass concentrations and absorption coefficients (factor 5 to 10 outside the 95th percentile of the other samples), which indicates a possible contamination from local sources. The measured

refractory BC mass concentrations of the aerosolized snow samples were corrected for the Marin-5 nebulizing efficiency to determine the BC mass concentrations per mL volume of melted snow. This BC concentration is shown in Figure 8c in conjunction with the eBC mass concentration of ambient air that is routinely measured by the German Federal Environment Agency using a Multi-Angle Absorption Photometer (MAAP). Also, a selection of meteorological data is presented in Figure 8 to highlight the trends in ambient temperature, sunshine duration, snow precipitation and snow height over the period the snow samples were collected at the UFS station. Although there is no clear correlation between the fresh snow samples and ambient air eBC mass concentration, the enhanced air eBC mass concentration observed end of March and beginning of April might have resulted in additional deposition of BC particles in the snow surface that is reflected - with a time lag of several days - in the measured snow refractory BC mass mixing ratio. Interestingly, this period of higher air eBC concentration is distinguished by a low precipitation activity, long sunshine periods as well as frequent daily maximum temperatures above the melting point that resulted in frequent thaw/freeze cycles and, consequently, a gradual decrease of the snow height by 30 to 40 cm. All in all, the enhanced air eBC concentration in conjunction with the meteorological conditions would favor enhanced BC mass concentrations in the fresh snow samples collected after precipitation events within this period or shortly after. Figure 9 shows corresponding mass size distributions of the refractory BC concentrations shown in Figure 8c averaged over the periods November to January, February and March, as well as April and May. For comparison purposes, the average size distributions are normalized by the corresponding total mass concentration $M_{total}$, which was deduced from a lognormal fit. The SP2-derived refractory BC mass size distribution only includes particles up to a mass-equivalent diameter of 560 nm, which means that larger BC particles are not recorded by the SP2. However, the average BC mass size distributions have distinct mode maxima at the mass median diameters (MMD) of 227, 194, and 222 nm for the Nov-Jan, Feb-Mar, and Apr-May periods, respectively. This indicates no strong seasonality in the snow BC mass size distribution even in the Apr-May period where the BC mass concentration in the snow was significantly enhanced (Figure **8**c). This further implies that indeed fresh snow was sampled which hasn't experienced thaw/freeze cycles severe enough to induce an agglomeration of the BC particles in the top snow layer. This conclusion is further supported by comparing the average BC mass size distributions of our snow samples with the BC mass size distribution of a fresh snow sample collected after a long-lasting snowfall event at Ny-Ålesund, Svalbard, Norway by Sinha et al. (2018) and with the averaged BC size distribution from five snow samples collected after three snowfall events in the semi-rural and rural surroundings of Denver, CO, USA by Schwarz et al. (2013). Our average fresh snow sample size distributions peak at similar MMD between 194 and 227 nm compared to the $223 \pm 28$ nm of the Sinha et al. study and the ~220 nm of the Schwarz et al. study. In addition, our size distributions indicate a non-lognormal shoulder at the upper size limit of the SP2 measurement that is in a very good agreement with the Schwarz et al. (2013) samples where the refractory BC mass size distributions were measured by a SP2 with modified detector gains up to 2 μm (see Figure 9). As pointed out by Schwarz et al. (2013) such snow BC mass size distributions reflect the typical atmospheric BC mass size distribution that is observed at remote locations altered by agglomeration and size selection processes during snow formation in the atmosphere. The good agreement between the mass size distributions of our snow samples and the average distribution of the Schwarz et al. (2013) samples allows us to estimate the refractory BC mass that is contained in the large particle size shoulder outside our measurement range. According to Schwarz et al. (2013) a fraction of 28% of the total BC mass can be attributed to particles with mass-equivalent diameters larger than 600 nm. A mass correction factor of 1.39 is therefore applied to the SP2-derived refractory BC snow concentrations in the following analysis.

For the assessment of the albedo effect of particulate impurities in snow surfaces the spectral absorption that is contained in the snow has to be quantified. As already mentioned in the introduction this is usually achieved by quantifying the mass mixing ratio of light absorbing particles in the snow and applying a mass specific absorption cross section to that particle mass resulting in a total absorption cross section per snow mass $\sigma_{abs}$ (given in m²/mL). With the measurement setup given in Figure **1**, this quantity is directly assessible. To deduce $\sigma_{abs}$ the measured absorption coefficient $b_{abs}$ [m⁻¹] of the aerosol released from the snow sample is converted by following equation

$$\sigma_{abs} = 10^{-3} \cdot b_{abs} \cdot R_{neb}/R_{pp} \cdot \varepsilon_{neb}^{-1} , \tag{1}$$

with $R_{neb}$ and $R_{pp}$ the air and liquid sample flow rates of the nebulizer, and $\varepsilon_{neb}$ the nebulizing efficiency. A unit conversion factor of $10^{-3}$ is necessary because $b_{abs}$ is given as [m²/m³], while $R_{neb}$ is given as [L/min]. $\sigma_{abs}$ values are calculated for the 31 UFS snow samples using Eq. (1) and are plotted in Figure 10 as a function of the corresponding refractory BC mass concentrations $c_{BC}^{SP2}$, which were corrected for the missing larger particle mass in accordance with the discussion above. This results in a good correlation between the snow mass specific absorption cross section $\sigma_{abs}$ and the refractory BC mass concentration of the snow samples with the mass absorption cross section, MAC, of the snow particles as correlation factor. In Table 1, the MAC values of the snow samples are compared with those determined for the Fullerene soot suspension standards. It is clear from Table 1 that the MAC of the snow particles is significantly larger than the MAC of the Fullerene soot with wavelength-dependent "enhancement" factors of 2.0, 1.9, and 1.4 for 405, 532, and 658 nm, respectively. This observation suggests that (*i*) the BC particles in the snow are thickly coated with transparent or low absorbing material that results in a real absorption amplification of the internally mixed particles by the so called lensing effect (e.g. Schnaiter et al., 2005), and/or (*ii*) part of the absorbing aerosol mass in these samples might be mineral dust or brown carbon that is co-deposited with the BC mass and that has a significant and strong wavelength-dependent mass absorption cross section in the visible spectral region (Schnaiter et al., 2006; Wagner et al., 2012). Both explanations are conclusive for atmospheric aerosol observed at a remote location like the UFS. Although the wavelength-dependence of the observed absorption "enhancement" suggests an insignificant impact from thickly coated BC particles – as this should show a larger absorption amplification in the red compared to the blue spectral range (Schnaiter et al., 2005) – the lensing effect is strongly depending on the actual coating thickness, the coating material, the composite particle size, and the geometrical particle configuration (Kahnert et al., 2012; You et al., 2016). Further, the mean AAE of the snow samples for the spectral ranges from 405 to 658 nm and 532 to 658 nm is $1.20 \pm 0.85$ and $2.10 \pm 2.24$, respectively, which is significantly larger but more varying than $0.46 \pm 0.07$ and $0.60 \pm 0.12$ deduced for the Fullerene soot suspensions for the same spectral ranges (Table 1). This suggests that it is more likely that the BC particles in the snow are accompanied by non-BC aerosol particles in varying amounts that induce additional absorption predominantly in the blue and green part of the visible spectrum.

Doherty et al. (2010) analyzed spectroscopic measurements of Arctic snow samples using the ISSW photometer to deduce the equivalent BC mass concentration $c_{BC}^{equiv}$, i.e. the amount of BC that would need to be present in the snow to account for the measured absorption. With the concurrent PAAS-3λ and SP2 measurements presented here, this quantity can be deduced in a similar way for the fresh snow samples from the UFS. For this purpose, the Fullerene soot absorption-equivalent mass concentration $c_{BC}^{equiv}$ was determined from the snow mass specific absorption cross section $\sigma_{abs}$ (Eq. 1) by applying the $MAC_{FS}$ determined for the Fullerene soot suspensions (Sect. 4 and Figure 7)

$$c_{BC}^{equiv} = \sigma_{abs}/\text{MAC}_{FS} .$$ (2)

In Figure 11, the deduced $c_{BC}^{equiv}$ values for the 31 snow samples are plotted as a function of the $c_{BC}^{SP2}$ mass concentrations. Interestingly, the two quantities are well correlated ($R^2$ between 0.89 and 0.93) with correlation coefficients $\gamma = \Delta c_{BC}^{equiv}/\Delta c_{BC}^{SP2}$ of 2.0, 1.9, and 1.4 for the 405, 532, and 658 nm wavelength, respectively, i.e. substantially different from unity. This indicates that (*i*) there is a significant fraction of light absorption in the

particle mass that cannot be attributed to refractory BC, (*ii*) the reason for this additional absorption is correlated with the BC mass, and (*iii*) the additional absorption has a strong wavelength dependence between the blue and red part of the visible spectrum. To further elaborate this observation the snow mass specific absorption cross section of the non-BC particles, $\sigma_{abs}^{nonBC}$, was calculated from $\sigma_{abs}$ of all particles

$$\sigma_{abs}^{nonBC} = \sigma_{abs} - c_{BC}^{SP2} \cdot \text{MAC}_{FS} \cdot 10^{-9} ,$$ (3)

with $\text{MAC}_{FS}$ the mass absorption cross section of Fullerene soot and a conversion factor of $10^9 [\text{ng/g}]$. Figure 12 shows the statistical analysis of $\sigma_{abs}^{nonBC}$ for the 31 snow samples of the present study. Thus, the non-BC particles show an absorption characteristic with a gradual increase of $\sigma_{abs}^{nonBC}$ with decreasing wavelength, which is accompanied by a strong increase of its variability. Again, this points to co-deposited aerosol mass that predominantly absorbs in the blue and green part of the visible spectrum. As already mentioned, possible

candidates for this additional light absorption are mineral dust and BrC.

Saharan dust events are routinely monitored by DWD based on a combination of particle size distribution and calcium ($Ca^{2+}$) concentration measurements, which defines the Saharan Dust Index (SDI; Flentje et al., 2015). Based on the latest SDI inventory[1], the UFS station was influenced by Saharan dust on approximately 20 days within the period January to May 2017. Therefore, it is conclusive that Saharan dust likely influences the light

absorption of the UFS snow samples. In an aerosol chamber study Wagner et al. (2012) deduced the complex refractive index, $m = n + ik$, of Saharan soil dust samples collected in a source region in southern Morocco during the SAMUM-1 field project (Heintzenberg, 2009). In Figure 12, $\sigma_{abs}^{nonBC}$ is compared with the average spectrum of the imaginary part, $k$, of the refractive index deduced for the three Moroccan dust samples of the Wagner et al. (2012) study. Such a comparison is reasonable as the absorption cross section of mineral dust is dominated by the

imaginary part of the refractive index (as well as the particle size distribution) and less by the real part. While the Saharan dust spectrum closely resembles the spectral signature of $\sigma_{abs}^{nonBC}$ with a very good match of the average values, the low spectral resolution and the high statistical variation of the $\sigma_{abs}^{nonBC}$ data might also allow for a different interpretation. Schnaiter et al. (2006) used a propane diffusion flame to generate carbonaceous aerosol particles with different organic carbon (OC) mass fractions in the range from about 10 to 70%. They found a strong

correlation between the OC mass fraction and the wavelength dependence of the aerosol absorption with AAE between 1 and as large as 9. Consequently, the particulate combustion emissions had different colors from black to brown to yellow, therefore representing brown carbon aerosol. Two examples from the Schnaiter et al. (2006) study are shown in Figure 12 to highlight a possible contribution of BrC to the non-BC absorbing aerosol mass in the snow samples. These two examples with OC mass fractions of 30 and 50% and mass specific absorption cross

sections of 3.8±0.5 and 1.4±0.5 m²/g, respectively, are capable to cover the short-wavelength variation in $\sigma_{abs}^{nonBC}$

---

[1] https://www.dwd.de/EN/research/observing_atmosphere/composition_atmosphere/aerosol/cont_nav/saharan_dust.html

observed for the UFS snow samples. Here, a reasonable mass concentration of 4 and 18 ng/mL was assumed for the 30 and 50% OC sample, respectively, to calculate the snow mass specific absorption cross section of BrC, $\sigma_{abs}^{BrC}$, from the corresponding MAC that is given in Schnaiter et al. (2006). In summary, from a spectroscopic perspective the additional light absorbing particle mass observed in the fresh UFS snow samples can be explained

by long-range transported Saharan desert dust and/or BrC particles that are co-deposited in the snow together with the BC particles in varying compositions and mass concentrations.

To further examine the nature of the particulate components that are deposited in the snow samples ion chromatography (IC) and an inductive coupled plasma mass spectrometry (ICP-MS) analysis were exemplarily conducted for the snow sample from March 10, 2017, mainly to clarify the concentration of higher ions which

might be present in the snow. Additionally, the aerosol released from this snow sample was fed to a Waveband Integrated Bioaerosol Sensor (University of Hertfordshire, UK, WIBS4) to get information on the biogenic particle fraction. For these additional analyses, re-aerosolized airborne particles were sampled downstream the nebulizing system behind the dryer by substituting the PAAS-3λ and the SP2 (see Figure 1). A Nuclepore™ filter with pore sizes of 0.2 µm was taken for Environmental Scanning Electron Microscopy (ESEM; Quattro S, ThermoFisher

Scientific, USA) microscopy combined with EDX microanalysis (EDAX, Octane Elite Super) to further characterize the different particle types found mainly in the larger particle size range (larger than ~500 nm) of this sample. The snow sample from March 10, 2017 has a mass concentration $c_{BC}^{SP2}$=2.8 ng/mL and an equivalent BC mass concentration of $c_{BC}^{equiv}$=6.0 ng/ml for λ=532 nm, which gives an enhancement factor of $\gamma = 2.1$. Therefore, the March 10 sample represents the bulk of the samples in terms of $\gamma$, but is on the lower side concerning $c_{BC}^{SP2}$ and

$c_{BC}^{equiv}$ concentrations (see Figure 11). The IC analysis of the snow sample, which was set to detect anions, shows only little concentrations of chloride, nitrate and sulphate of 0.29, 1.1 and 0.3 mg/L, respectively. Only very low concentrations of alkaline and alkaline earth metals were found from the ICP-MS analysis. For the trace metals manganese, iron, copper and zinc concentrations of 9.7 µg/L, 1.7 µg/L, 1.1µg/L and 8.7µg/L were found, respectively.

The ESEM micrographs reveal that the larger (>~500 nm) particles extracted from the March 10 snow sample predominantly consist of biogenic and biological materials including fragments of cellular membranes, whole bacteria, pollen, spores, and their mixtures. Mineral dust particles could be identified in the sample too, but to a much less extent than the biogenic particles. Figure 13 gives an overview composite image of a typical Nuclepore™ filter area, where particles with heavier elements like Al, Si, Fe, Mg, K, and Ti, are accentuated in

green color due to their brighter response in the backscatter electron detector (BSED). These elements are typically found in mineral dust particles, as compared to the lighter elements like C, N, O, Na, and S typical for biological material. This overview picture highlights the low relative abundance of mineral dust particles in the coarse mode particle size range of the sample. Representative examples of individual particles are given in Figure 14 Figure 16. Note that the EDX spectra of all analyzed particles are very characteristic for particle agglomerates or for chemical

aging. The biogenic particle (Figure 14) has areas showing intracellular composition (spot A) and pure cellular membrane fragments (spot B), whereas the mineral dust particle (Figure 15) and soot particle (Figure 16) exhibit spectra characteristics for both inorganic and biogenic material. An example of a Fullerene soot aggregate, emitted from one of the aqueous suspension standards is shown in Figure 17. As expected, the Fullerene soot particle does not contain any foreign chemical elements, as shown by the EDX spectrum in the right panel of Figure 17.

The WIBS4 discriminates fluorescing biological aerosol particles (FBAP) by combining single particle fluorescence signals from two excitation/emission wavebands with a low cross-sensitivity to inorganic, combustion, and mineral dust particles (Toprak and Schnaiter, 2013). The WIBS4 measurement of the March 10, 2017 snow sample supports the ESEM results of a high fraction of biogenic particles (43%) in the size range larger than 0.5 μm. The size segregated analysis reveals biogenic particle fractions of 80% and 100% for sizes larger than 2 μm and 3 μm, respectively.

While these results give a detailed look into the physical and chemical nature of the of the particles that might contribute the light absorption in the March 10 snow sample, they cannot used to draw conclusions for all snow samples. Here, further analyses are required that couldn't conducted within the scope of this pilot study. However, one question that arises from the above findings is whether the biogenic particles found in the March 10 snow sample can be attributed to BrC, which was shown to be a good candidate for explaining the additional light absorption in the snow samples (Figure 12). The term "brown carbon" is mainly related to a strong wavelength dependence of the visible light absorption observed in these materials. From a chemical perspective, BrC can generally be divided into humic-like substances (HULIS) and tar balls (Wu et al., 2016). HULIS can be characterised mainly as a mixture of macromolecular organic compounds with various functional groups and are expected e.g. in oxidation processes of biogenic precursors (Wu et al., 2016). Tar balls are emitted from biomass burning and are of spherical, amorphous structure and are typically not aggregated. Moreover, light absorbing organic material and HULIS can be formed from the water-soluble fraction of biomass burning aerosol compounds, and is therefore suggested as an atmospheric process for the formation of light absorbing BrC in cloud droplets (Hoffer et al., 2004). Further examination of snow samples from different locations as well as systematic investigations on the optical behaviour of biogenic particulate matter is therefore necessary to evaluate the influence of biogenic (including biological), BrC and mineral dust on the aerosol absorption properties in the visible spectral range.

## 6. Conclusions

In this study a new laboratory analysis method for snow and ice samples was presented. With this method the snow mass specific absorption cross section $\sigma_{abs}$ is directly measured by the three-wavelength photoacoustic absorption spectrometer PAAS-3λ on re-aerosolized snow samples without particle deposition on filters. The refractory black carbon (BC) mass concentration in the snow samples was concurrently determined using a single particle soot photometer (SP2). Using water suspensions of Fullerene soot particles of known BC mass concentrations as a BC reference for the snow samples, the aerosolization efficiency of the nebulizer was quantified and the detection limit of the method was assessed. Further, the mass specific absorption cross section of Fullerene soot (MAC$_{FS}$) was determined for the visible spectral range from the concurrent PAAS-3λ and SP2 measurements.

The method was then used to analyse 31 fresh snow samples collected at the Environmental Research Station Schneefernerhaus (UFS) in the winter period 2016/2017. The spectral snow mass specific absorption cross sections $\sigma_{abs}$ measured by the PAAS-3λ were analysed as a function of the refractory BC snow mass mixing ratio $c_{BC}^{SP2}$ deduced by the SP2 to determine the BC mass specific absorption cross section (MAC) and the equivalent BC mass mixing ratio $c_{BC}^{equiv}$ of the snow samples. Contrasting the MAC of the snow samples with the MAC$_{FS}$ of the Fullerene soot reference BC material, it was found that the MAC of the snow particles was enhanced by a factor

of two in the blue and green part of the visible spectrum, resulting in an enhanced $c_{BC}^{equiv}$ mass mixing ratio compared to $c_{BC}^{SP2}$. While the latter accounts only for the refractory BC it was concluded that the discrepancy between the optically deduced $c_{BC}^{equiv}$ and the $c_{BC}^{SP2}$ suggests the presence of light absorbing non-BC particles in the snow samples. The good correlation between $c_{BC}^{equiv}$ and $c_{BC}^{SP2}$ further indicates that the non-BC and BC aerosol particles have either the same source (e.g. biomass burning) or experienced a significant atmospheric processing (internal mixing) before they were deposited into the snow. Using the $MAC_{FS}$ of Fullerene soot and the $c_{BC}^{SP2}$ mass mixing ratio measured for the snow samples, the snow mass specific absorption cross section $\sigma_{abs}^{nonBC}$ of the non-BC particles could be determined. The spectral behaviour of $\sigma_{abs}^{nonBC}$ gives mean absorption Angström exponents of 2.2 and 1.5 for the 405 to 532 nm and 532 to 658 nm spectral ranges, respectively, indicating that the non-BC light absorbing particle mass is predominantly absorbing in the blue to green part of the visible spectrum and less in the red. By comparing $\sigma_{abs}^{nonBC}$ with laboratory data for Saharan dust and organic (brown) carbon, it could be shown that these atmospheric aerosol components can explain the observed non-BC light absorption in the snow. Additional analyses of an exemplary snow sample using environmental scanning electron microscopy combined with EDX microanalysis as well as single particle fluorescence measurements revealed that the larger particles of the snow sample are predominantly of biogenic or organic origin with lower contributions from mineral dust. This finding supports the above interpretation that the additional non-BC light absorbing aerosol mass is likely due to biogenic particles, brown carbon, and mineral dust. However, further studies are required including samples from other locations, to quantify the general contribution of these non-BC atmospheric aerosol components to the visible light absorption in snow and ice surfaces and the resulting albedo reduction.

## 7. Acknowledgements

We thank Ludwig Ries, German Federal Environment Agency (UBA) and Gerhard Müller, German Meteorological Service (DWD), Global Observatory Zugspitze/Hohenpeissenberg for providing weather, atmospheric BC and Saharan dust inventory data. Joshua Schwarz, National Oceanic and Atmospheric Administration (NOAA), Boulder, USA is thanked for providing snow sample data for comparison. This work was funded within the Helmholtz Research Program Atmosphere and Climate.

## 8. Tables

Table 1: Overview of the optical and BC mass properties measured for the Fullerene soot suspension standards as well as the fresh snow samples.

| | Fullerene Soot | UFS Snow Samples |
|---|---|---|
| MAC (405 nm) [m²/g]<br>(mean ± 1σ)<br>(5 - 95th percentile range) | 10.5 ± 3.2<br>7.6 - 14.0 | 21.1 ± 7.9<br>13.4, 33.1 |
| MAC (532 nm) [m²/g]<br>(mean ± 1σ)<br>(5 - 95th percentile range) | 9.5 ± 2.2<br>7.7 - 16.0 | 18.2 ± 7.2<br>10.9, 27.2 |
| MAC (658 nm) [m²/g]<br>(mean ± 1σ)<br>(5 - 95th percentile range) | 8.6 ± 3.3<br>5.7, 10.9 | 11.9 ± 4.6<br>8.1, 20.2 |
| AAE (405 - 658 nm)<br>AAE (405 - 532 nm)<br>AAE (532 - 658 nm)<br>(mean ± 1σ) | 0.46 ± 0.07<br>0.35 ± 0.08<br>0.60 ± 0.12 | 1.20 ± 0.85<br>0.49 ± 1.00<br>2.10 ± 2.24 |
| Mass Median Diameter [nm]<br>(mean ± 1σ)<br>(5 - 95th percentile range) | 228 ± 3<br>223.5 - 233 | 207.5 ± 42.1<br>146.2 - 290.4 |
| 1σ Distribution Width<br>(mean ± c)<br>(5 - 95th percentile range) | 1.57 ± 0.018<br>1.54 - 1.6 | 1.83 ± 0.13<br>1.64 - 2.06 |
| $\sigma_{abs}$ (405 nm) [$10^{-8}$ m²/mL]<br>(median, 5 - 95th percentile range) | | 9.9, 4.3 - 37.9 |
| $\sigma_{abs}$ (532 nm) [$10^{-8}$ m²/mL]<br>(median, 5 - 95th percentile range) | | 8.4, 2.9 - 34.2 |
| $\sigma_{abs}$ (658 nm) [$10^{-8}$ m²/mL]<br>(median, 5 - 95th percentile range) | | 5.9, 1.9 - 21.5 |
| $c_{BC}^{SP2}$ [ng/mL]<br>(median, 5 - 95th percentile range) | | 5.0, 1.7 - 20.4 |
| $c_{BC}^{equi}$ (532 nm) [ng/mL]<br>(median, 5 - 95th percentile range) | | 8.9, 3.1 - 36.1 |

**9. Figures**


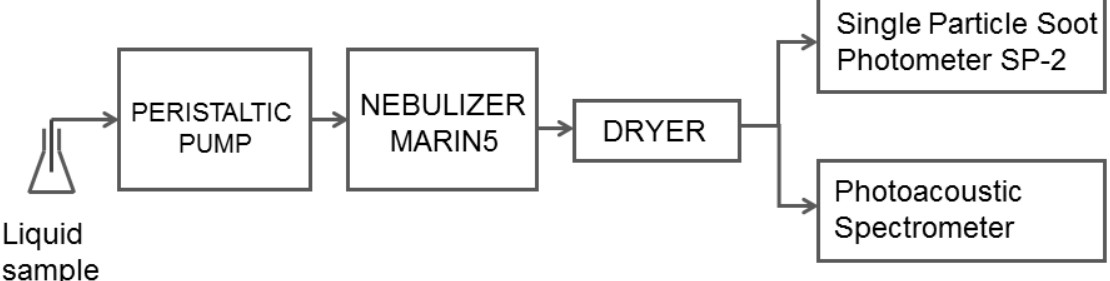

Figure 1: Schematic of the instrumental setup used in the present study.


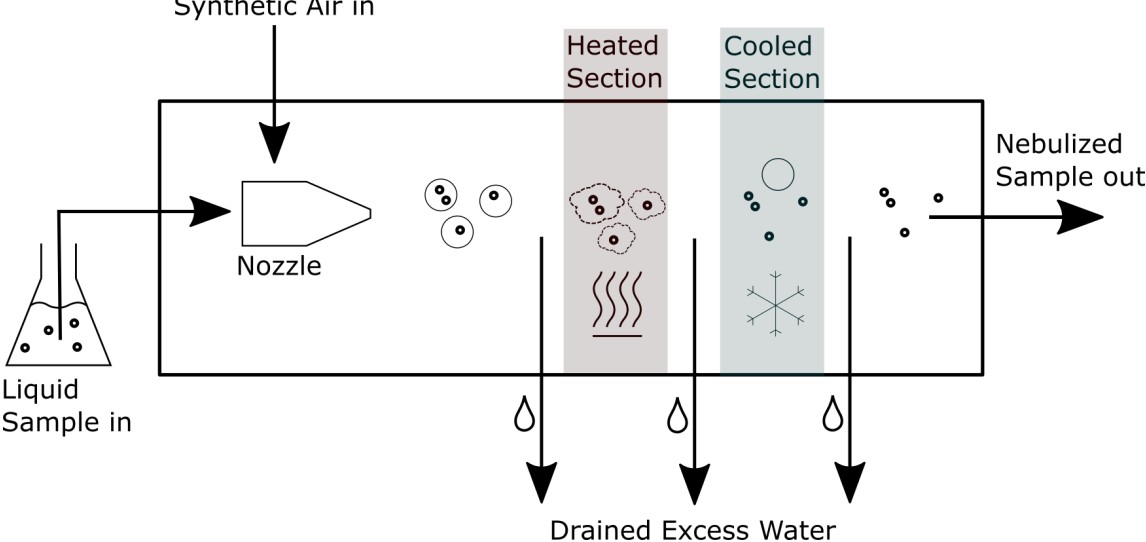

Figure 2: Operation principle of the CETAC Marin-5 nebulizer used in the present study.

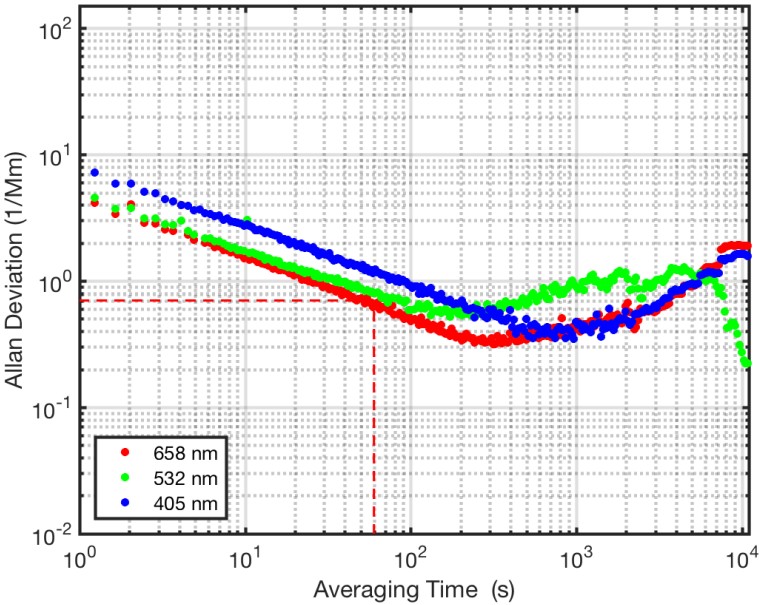


Figure 3: Allan deviation analysis of a long-term photoacoustic background measurement. The red dashed lines indicate the typical averaging time of 60 s used in the PAAS-3λ measurements and the corresponding Allan deviation (1σ) of 0.7 1/Mm.

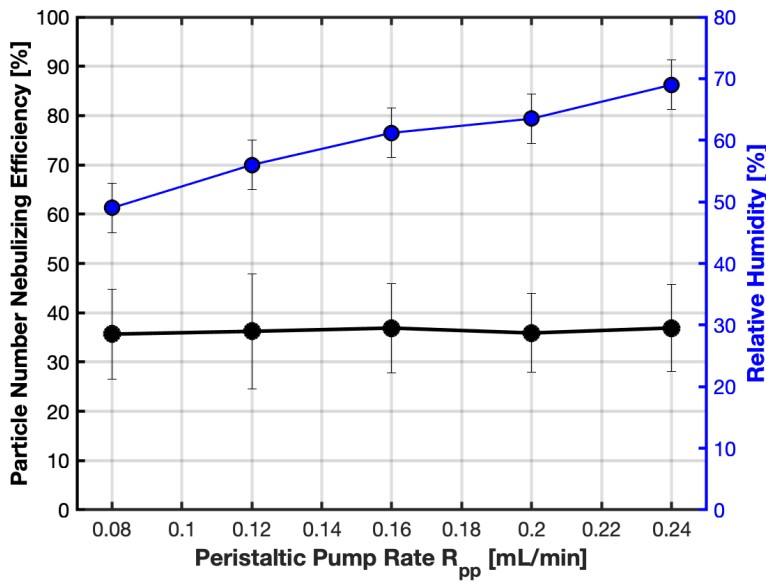


Figure 4: Particle number nebulizing efficiency and relative humidity of the output aerosol flow as a function of the peristaltic pump flow rate  applied in Marin-5 nebulizing tests with PSL particle suspensions.

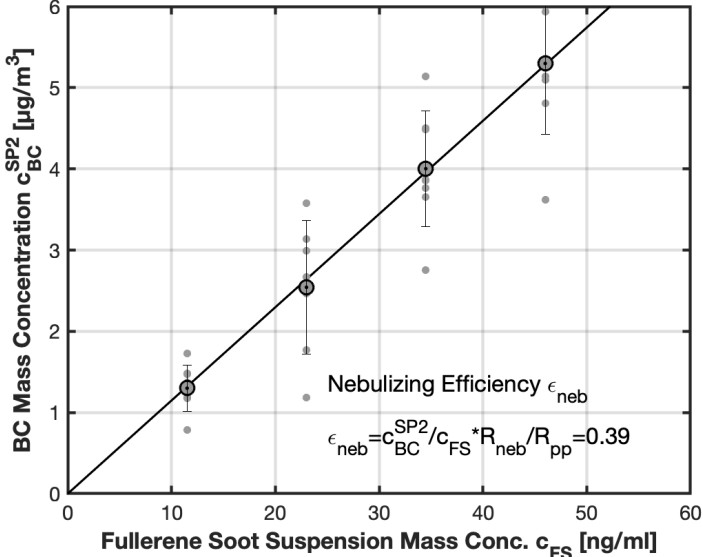

Figure 5: Particle mass nebulizing efficiency  of the Marin-5 nebulizer using Fullerene soot suspensions with
defined BC mass concentrations $c_{FS}$. The refractory BC mass concentration $c_{BC}^{SP2}$ was measured in the dry
aerosol exit flow of the nebulizer using a Single Particle Soot Photometer (SP2). $R_{neb}$ and $R_{pp}$ define the flow
rates of the dispersing gas and the liquid sample supply, respectively. See text for details.


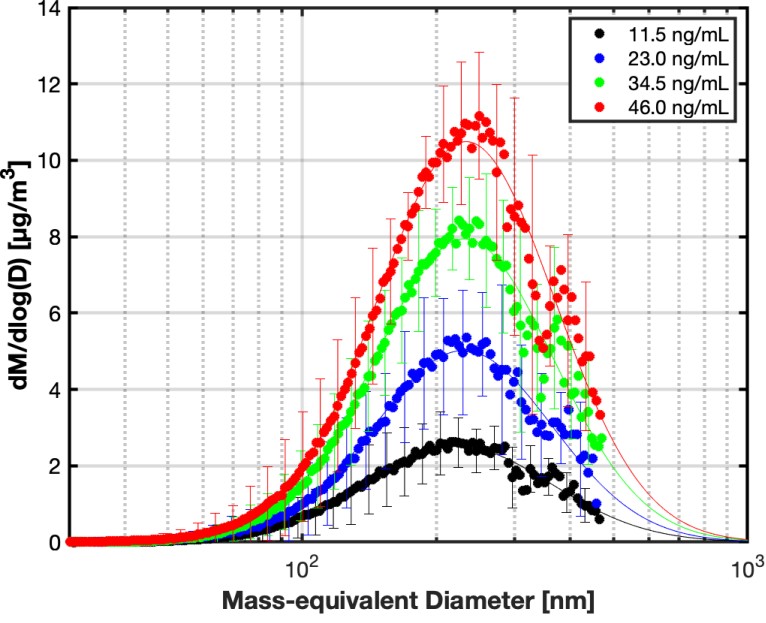

Figure 6: Particle mass size distributions of the Fullerene soot suspensions used to characterize the Marin-5
nebulizer. The measurements were conducted with the SP2, which has an upper size limit of 560 nm in the
present study. Results from lognormal fits are represented by the thin lines.

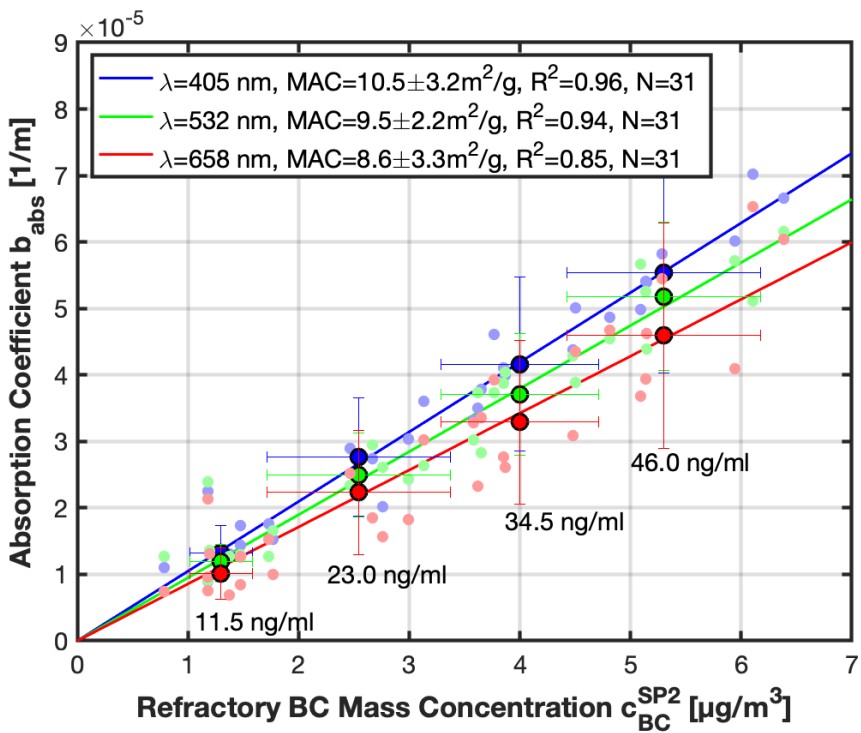


Figure 7: Determination of the mass specific absorption cross section (MAC) of re-aerosolized Fullerene soot suspension standards at 405, 532, and 658 nm. The MAC values are given in the legend and are derived from concurrent measurements of the absorption coefficient, $b_{abs}$, using the photoacoustic aerosol absorption spectrometer PAAS-3λ and the refractory BC mass concentration, $c_{BC}^{SP2}$, using the SP2.



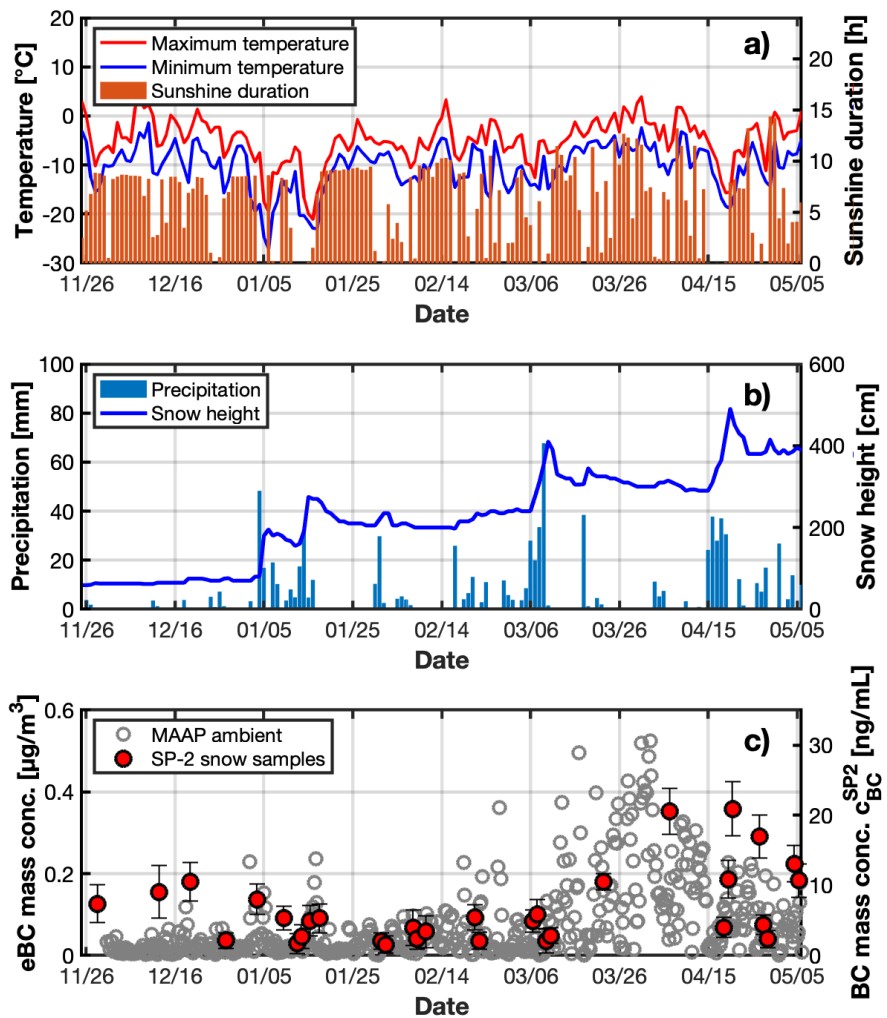

Figure 8: Overview of the meteorological and ambient BC conditions during the period the snow samples were collected at UFS. The data in (a) to (b) are on a daily basis. The atmospheric equivalent black carbon (eBC) mass concentrations shown in panel (c) represent 30 min averages of the MAAP measurements. Note that the refractory black carbon mass concentrations, $c_{BC}^{SP2}$, deduced from the SP2 measurements of the re-aerosolized snow samples are compared in panel (c) with the atmospheric eBC mass concentration. See text for details concerning data providers, sampling, and measurement methods.


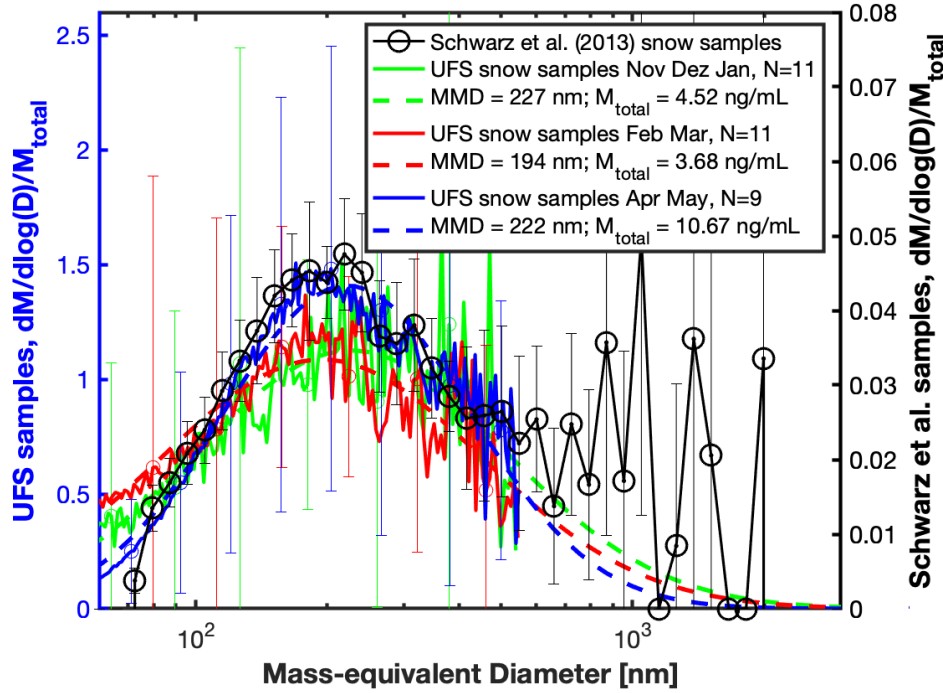


Figure 9: BC mass size distributions of the snow samples deduced from the SP2 measurements. The size distributions are averaged over the three periods Nov-Dec-Jan (green), Feb-Mar (red), and Apr-May (blue) and are normalized by the total mass ($M_{total}$). Lognormal fits are represented by the dashed lines. Fit results in terms of mass median diameter (MMD) and integrated mass ($M_{total}$) are given in the legend. An averaged size distribution for fresh snow samples published by Schwarz et al. (2013) is shown for comparison (black line and open circles)


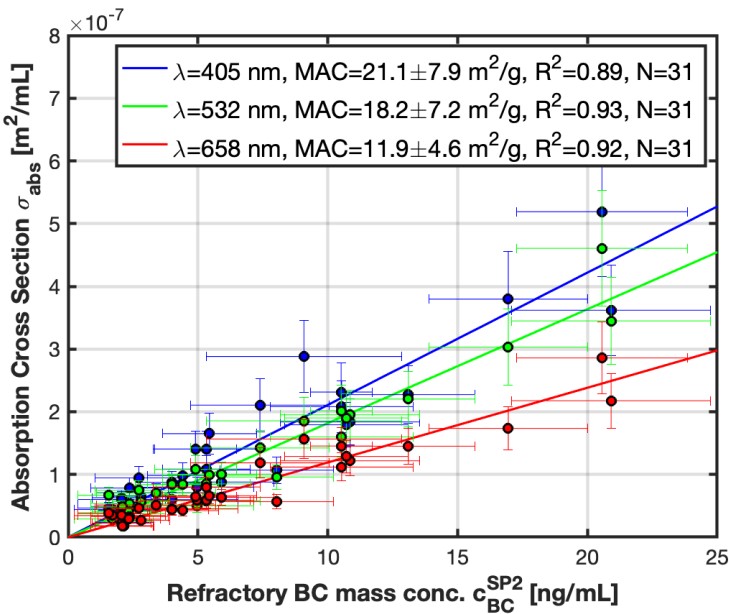

Figure 10: Snow mass specific absorption cross section $\sigma_{abs}$ of the snow samples (Eq. 1) as a function of the mass concentration $c_{BC}^{SP2}$ deduced from the SP2 measurements. The BC mass specific absorption cross section, MAC, is deduced from a linear regression fit of the data per wavelength and is given in the legend. See text for details.


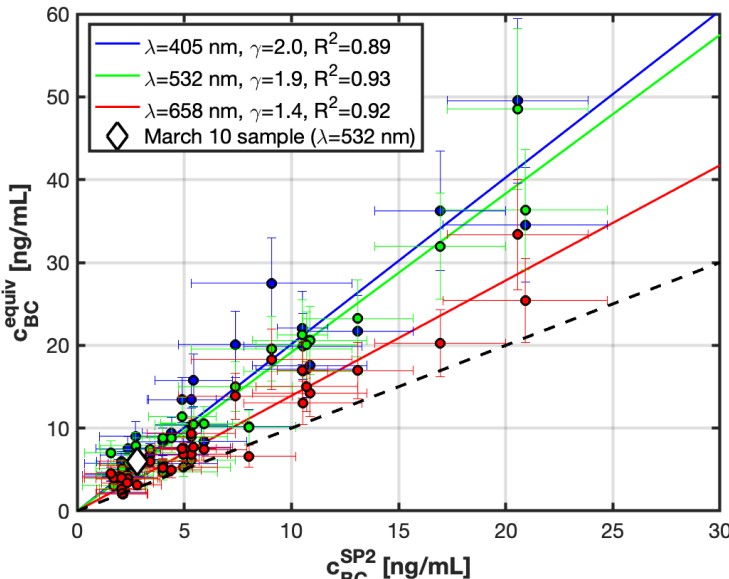


Figure 11: Equivalent BC mass concentration $c_{BC}^{equiv}$, Eq. (2), of the snow samples as a function of the refractory BC mass concentration $c_{BC}^{SP2}$. The dashed black line represents the 1:1 line. Linear regression fits per wavelength gives the mass "enhancement" factor $\gamma$, which is given in the legend. The white diamond symbol marks the 532 nm values of the March 10 snow sample that was further analyzed for elemental composition, particle morphology, and fluorescence response. See text for details.


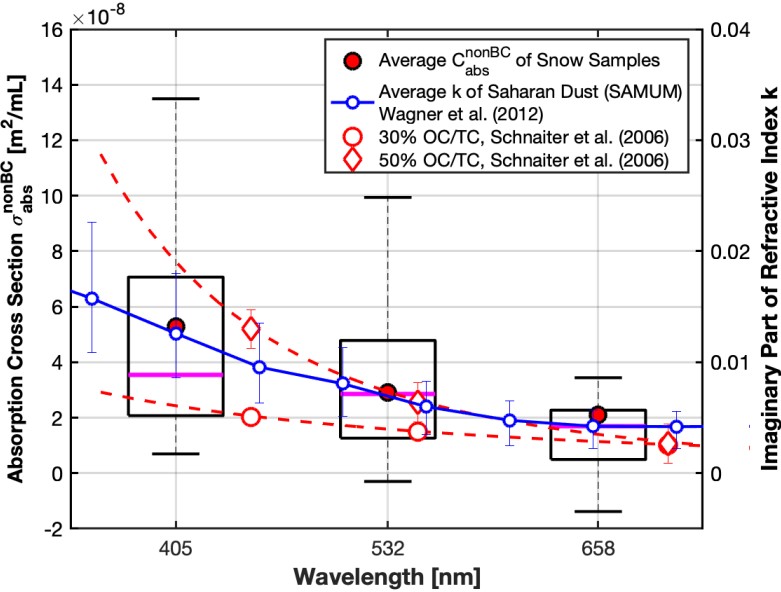

Figure 12: Statistical analysis of the snow mass specific absorption cross section $\sigma_{abs}^{nonBC}$ of the non-BC paricles deduced from Eq. (3). Laboratory data for Saharan dust (blue) and brown carbon (red) are shown for comparison. Two examples of brown carbon (BrC) with organic to total carbon mass ratio, OC/TC, of 30% and 50% are selected to emphasize the possible variability in spectral absorption of this class of atmospheric aerosol mass. A BrC in snow mass concentration of 4 and 18 ng/mL was assumed for the 30 and 50% example, respectively.


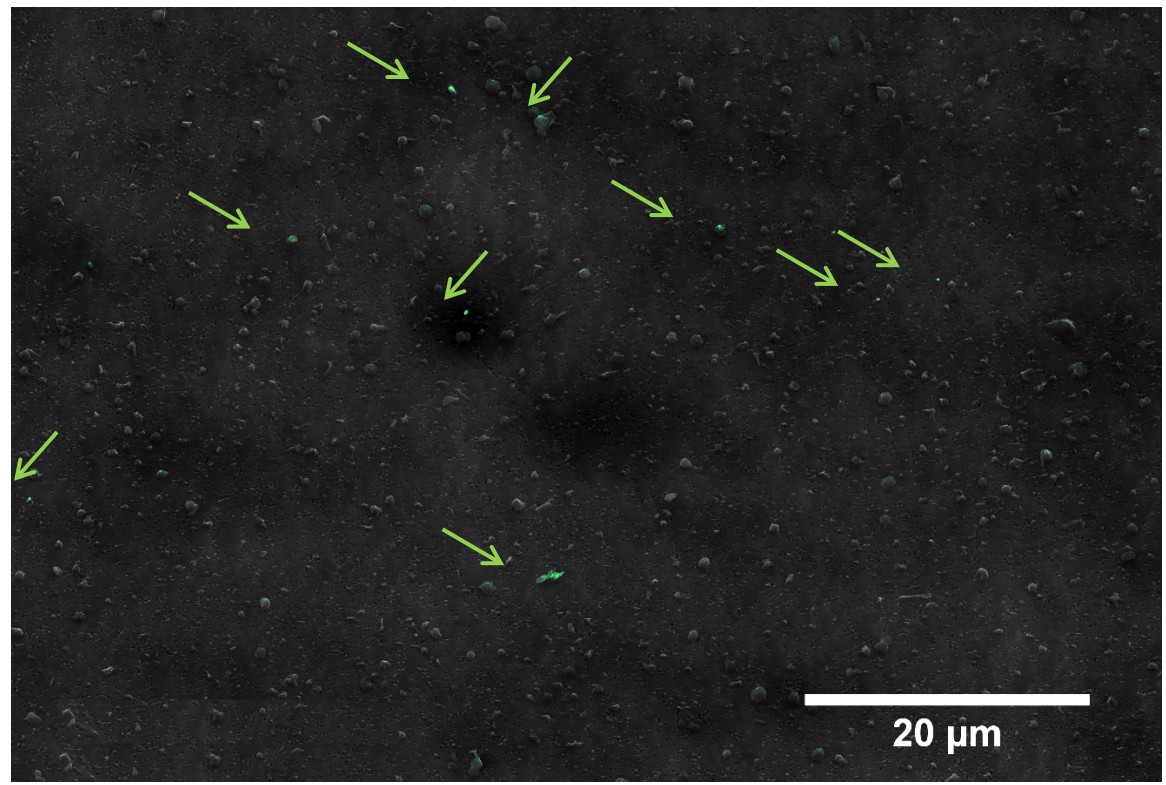

Figure 13: Overview of residual particles extracted from a snow sample via Marin-5. The image is a false color overlay from secondary electron (SE) detector (black and white image) and backscatter electron detector, BSED (the green spots). The particles containing elements with higher atomic number are visible as green spots and are highlighted by green arrows.

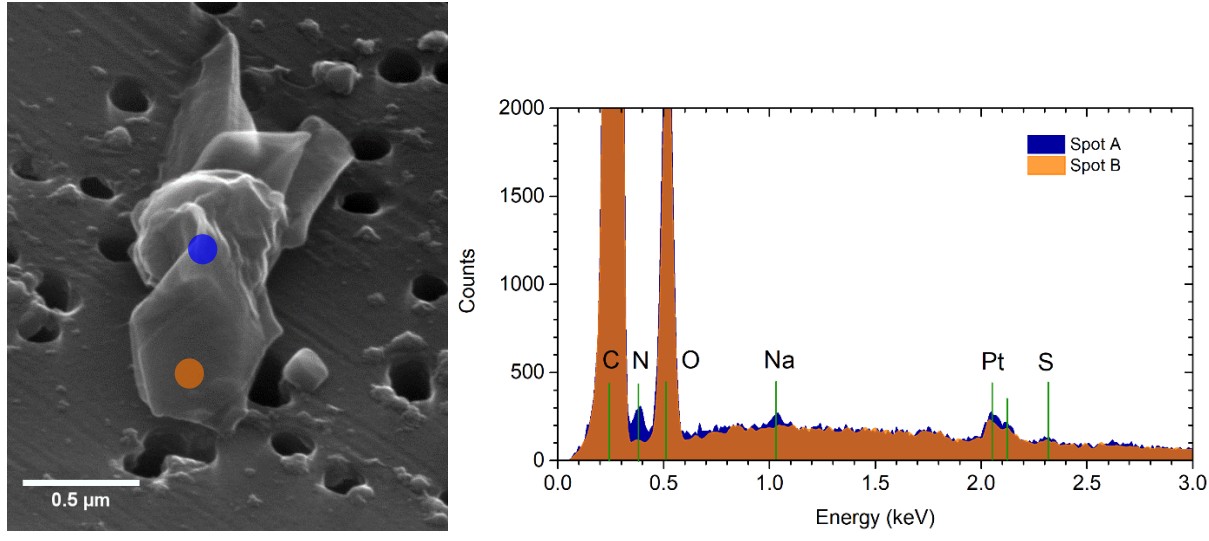


Figure 14: Morphology and composition of biological residual particle extracted from snow sample behind Marin-5 on a Nuclepore™ filter. Left panel: the SEM secondary electron image. Right panel: EDX-spectrum obtained from different areas of the particle (marked as color spots in the image). A clear biogenic signature (N, S, Na) is visible for the central bulky part suggesting intracellular composition (spot A, blue color), whereas the exterior part of the particle shows pure carbonaceous compounds (C, O) (spot B, orange color). Sample coating is responsible for the platinum peak.

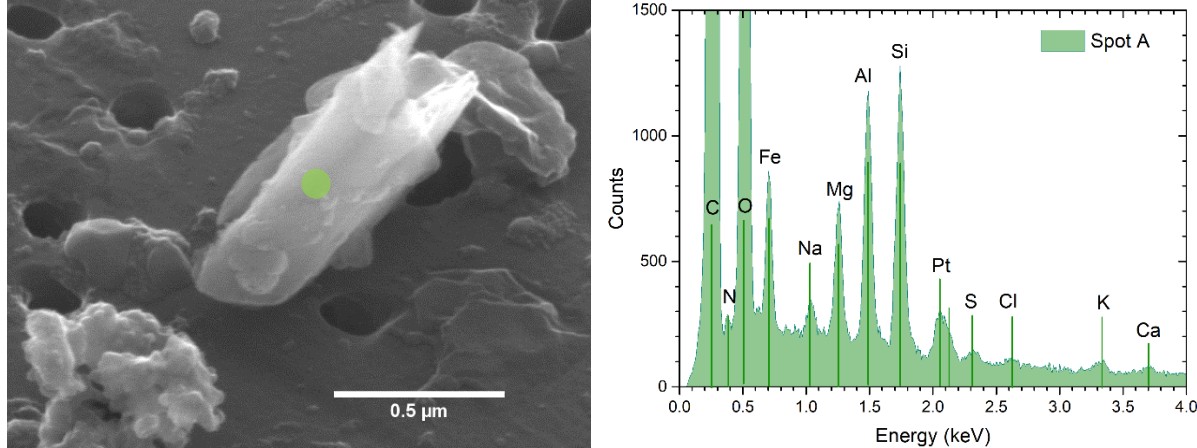

Figure 15: Same as Fig. 14 but for a mineral dust particle. The EDX-spectrum of the particle identifies chemical patterns that are characteristic for mineral dust (Al, Si, Mg, Fe, K, Ca), biogenic (N, Na, Cl, S), and carbonaceous (C, O) materials.


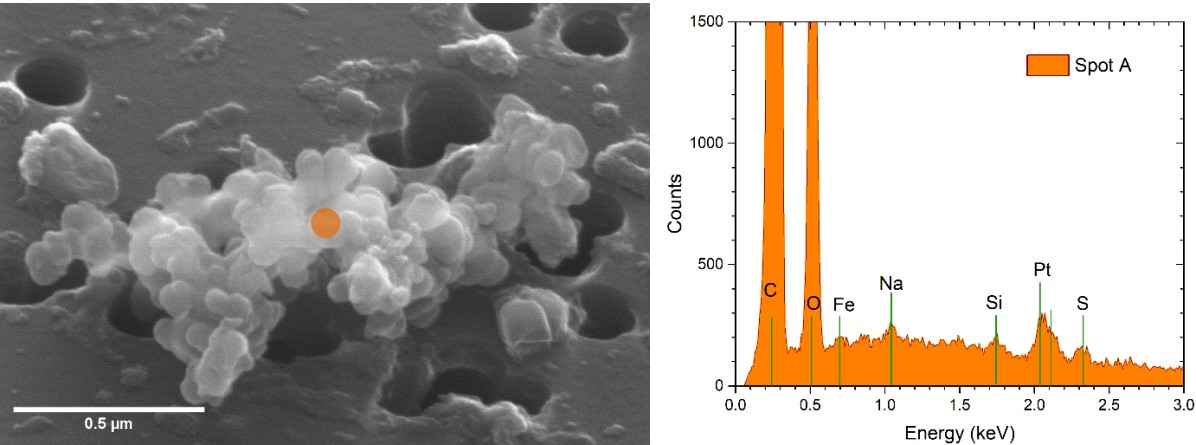

Figure 16: Same as Fig. 14 but for a soot (BC) particle. The EDX-spectrum of the particle reveals trace elements of Fe, Na, Si, and S in addition to the dominating C, O pattern that is characteristic for carbonaceous matter.


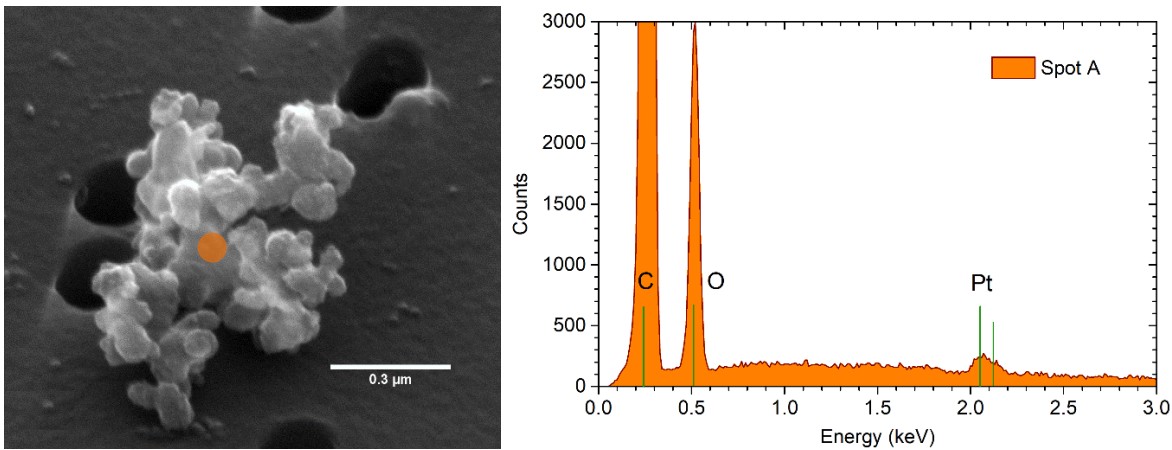

Figure 17: Same as Fig. 14 but for a Fullerene soot particle from the particle standard suspensions. The EDX-spectrum of the particle collected from the spot A shows a pure carbonaceous signature with only the C and O peaks. Sample coating is responsible for the Pt peak at 2.08 keV.

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
