# Peer review of "Specifying the light absorbing properties of aerosol particles in fresh snow samples, collected at the Environmental Research Station Schneefernerhaus (UFS), Zugspitze"

_Atmospheric Chemistry and Physics, 2018_

## Referee Comment (RC1) · Anonymous Referee #3 · 8 Feb 2019

Manuscript: Specifying light absorbing properties of aerosol particles in fresh snow samples, collected at the Environmental Research Station Schneefernerhaus (UFS), Zugspitze (Linke et al.,)

It is often regarded that the polar regions are the "canary in the coal mine" regarding this region's sensitivity to changes in radiative forcing. This sensitivity derives, in part, from the very high surface albedos that typify these regions. However, despite the recognized importance of this subject, our quantitative understanding of the radiative contribution by light absorbing aerosols - be it atmospheric or surface-deposited parti-

cles - in these regions - and on snow/ice - is still very low. Therefore, more research is indeed needed to improve our understanding of this subject and to reduce the associated uncertainty. To this end, the manuscript by Linke and co-workers undertake an investigation of light absorbing properties of aerosols deposited on snow near the Environmental Research Station Schneefernerhaus with the goal to better quantify the aerosol types (classes) of light absorbing aerosols present in this region. To derive the optical properties, these researchers combine aerosol light absorption measured by a photoacoustic spectrometer with refractory black carbon (rBC) mass loading measurements reported by a single particle soot photometer (SP2) on re-aerosolized snow samples. The insights and data that can be gleaned from a study such as this is of value and should, eventually, be published. But in its current state, the manuscript is not ready and thus is not recommended for publication at this time.

One of the biggest issues this reviewer has centers on how these researchers are combining their light absorption and rBC mass measurements. In the atmospheric aerosol community, we take these two datasets and derive the mass absorption cross-section (MAC; m2/g). This property enables researchers to say something about the mixing state of the rBC particles as well as something about aerosol type (e.g., the presence/absence of light absorbing organic material/particles (e.g., brown carbon or BrC and dust). However, despite the ubiquity of this methodology in our community, the authors, instead, advocate using a methodology that compares the SP2-derived mass with what they term as a "fullerene soot equivalent mass" derived from the photo acoustic spectrometer. In order to use a "fullerene equivalent mass" derived from the light absorption measurements the researchers must assume that the MAC is constant - the literature is populated with many studies that show that the MAC is not constant. For example, at 550 nm, the black carbon MAC for fresh (uncoated) soot is $\sim$ 7.5 m2/g while for thickly-coated black carbon particles this value could be 12-14 m2/g at this wavelength, thus representing a factor of 2x change. How do the authors account for the potential changes in the MAC brought about by changes in the rBC mixing state? Why not simply derive a MAC and utilize the variability in this value to infer

something about the light absorbing particles deposited on the snow? The authors provide no justification as to why their methodology is preferred. If the authors feel strongly that this methodology offers an advantage not available with a MAC-based methodology, then they need to present that argument. Further, any sources of error in the interpretation of the data using this methodology is not addressed and needs to be.

Continuing, this reviewer is very concerned about the reported MAC values for the fullerene standard. Fullerene soot should be uncoated and thus should exhibit a MAC that is characteristic of an uncoated black carbon particle, namely a MAC $\sim$ 7.5 m2/g at 550 nm. Yet, at 532 nm, the authors report a MAC of of 10.6 (+/- 2.8) m2/g - a value that is $\sim$40% that the canonical value for denuded soot, and 65% larger than the fullerene MAC reported by Zhou (6.4 m2/g) - work cited by the authors. Additionally, it is also at odds with the recently published value by Zangmeister 6.1 m2/g at 550 nm (Carbon (2018), doi: 10.1016/j.carbon.2018.04.057). The disagreement is quite critical as it will have bearing on interpretation of data collected on the snow samples. While the authors are correct that the BC MAC is dependent on particle size, using the argument that the differences between their higher values and that reported by others as being due to differences in the sample size distribution studied is not correct. Using a refractive index of 1.95-0.79i (Bond & Bergstrom (2006); Aero. Sci. Tech. 40:1, 27-67) a straight forward Mie calculation reveals that the maximum MAC calculated for a BC is $\sim$7.9 m2/g for a BC diameter of 150 nm and that the MAC only goes down with either increasing or decreasing particle size. Therefore another explanation is needed. Three potential explanations submitted here are: (i) either the fullerene soot is coated and the derived MACs reflect a lensing-enhanced value or (ii) there are some non-BC, light absorbing aerosols present in the sample, or (iii) the number concentrations used during the fullerene soot calibration are high enough that particle coincidence is occurring that the post-processing of the SP2 data does not correct for. This latter possibility is brought up because particle-resolved measurements, like the SP2, can easily find themselves in the particle-coincidence regime. Such an undercounting of

the actual fullerene mass would bring the reported MACs down and thus reconcile their MAC values with others. The authors are encouraged to insure that the SP2 is not operating in this regime.

While on this subject, what were the re-aerosolized number concentrations in the snow samples studied? On lines 243-244, the authors state that "….Marin-5 nebulizer was then fixed at a rate of 0.32 mL min-1, which guarantees a high enough particle mass concentration for the photo acoustic measurement. Depending upon the mass concentration and mode size of the particles, this could hint at co-incidence issues with the SP2.

Other specific issues:

Do the authors worry about the loss of water soluble BrC during sample preparation that might not be reflected when re-aerosolizing their snow samples?

Line 167: What were the PSL diameters used? While the incandescence channel of the SP2 is sensitive enough to detect <100 nm rBC particles, the scattering channel is typically limited to optical diameters > 200 nm. What is the particle number loading in the instruments (see reference to coincidence above)

Line 291-293: The authors report and discuss the "enhancement" factor of the soot-equivalent mass derived from the photoacoustic spectrometer with decreasing wavelength (1.65, 2.28, and 2.38 at 658 nm, 532, 405 nm, respectively) and conclude that this suggests these samples might contain mineral dust or BrC. The data might be able to say something more concrete. While the authors should conduct a more thorough literature search, mineral dust tends to absorb more in the red then the blue, whereas BrC exhibits the opposite wavelength dependence. The wavelength dependence of light absorption observed in the present study suggests that the non-rBC absorbing species is BrC. Also, it should be noted that BrC can exhibit absorption at the red wavelengths, though, again, the wavelength dependence favors more absorption in the blue (e.g., tar balls: Hoffer, et al., 2017. Brown carbon absorption in the red and

near infrared spectral region. Atmos. Meas. Tech. 10(6): 2353–59).

line 349: "The term brown carbon is not clearly defined or characterized." This is very misleading. Simply put, brown carbon, BrC, is organic aerosol that absorbs light. While the chemical composition for any given BrC aerosol may very, as long as the organic aerosol absorbs light it is cataloged as BrC. Please reword this sentence.

Please provide plot of fullerene standard size distribution.

Editorial comments

There is no reason to put quotes around the word fullerene.

Line 67: "Most Himalayan glaciers as glaciers elsewhere have retreated..." Please insert comma after first occurrence of the word "glaciers" and after "elsewhere".

Line 311: "To get a general idea of the nature of the components that are solved and dissolved within the snow samples ionic chromatography ..." the word "solved" does not seem right here. Please check and correct as necessary.

---

## Referee Comment (RC2) · Anonymous Referee #2 · 8 Feb 2019

Referee Comments for: "Specifying light absorbing properties of aerosol particles in fresh snow samples, collected at the Environmental Research Station Schneeferner-haus (UFS), Zugspitze"

General Comments:

This manuscript presents a novel method for concurrently measuring the rBC mass-concentration and total aerosol absorption in snow. The authors find a discrepancy between the SP2-measured rBC concentration and a calculation of rBC concentration

that is based on total aerosol absorption and a fullerene-standard's MAC. The former being smaller than the latter, the authors conclude the likely presence of non-rBC absorbing aerosols in the snow. WIBS measurements indicate that the larger of these non-rBC aerosol are predominantly of biological origin. These results are both interesting and important, as the presence and properties of light-absorbing aerosols in snow and their impact on snow/ice melt is not presently well understood.

Overall, the methodology of the measurements seems sound, and the text contains all of the necessary information to understand the experiment. I have four main concerns, three of which are science-related, and one regarding the writing. These can be found in the 'Specific Comments' below. I believe that once these things are appropriately addressed, the manuscript deserves publication in ACP.

Specific Comments:

My first concern is that this study's calculated MAC for fullerene soot is quite high compared to other literature, which typically put it around 7 m2/g or so. The authors acknowledge this, but I don't feel they explore (at least in the text) the underlying cause sufficiently. As this is at least somewhat dependent on the size of the fullerene particles, I suggest that the fullerene standard's size distributions be added to Figure 8.

Second, the authors don't estimate the percentage of rBC in snow that is above the detection limit of the SP2. Again, the authors acknowledge the issue, beginning on line 278, by stating that the SP2 only detects rBC up to 500 nm VED, but then rather simply declare the mass beyond this as small. The size distribution in Figure 8 indicates that there is still non-negligible mass above 500nm, and exactly how much can easily be estimated using a lognormal fit to the distributions. I attempted to estimate this by extracting data from the plot (see my attached figure), doing my own fit, and calculating percentage of area under the curve that is above 500nm...I get about 10%. Furthermore, its highly doubtful that the SP2 is detecting with 100% efficiency below 60-70nm or so. This probably has a smaller effect on the total rBC mass, but including

that data in a lognormal fit could skew the fit result. If I do the same fit, but only use the size. distribution data between 70 – 500 nm, I come up with ∼13% of the mass above the detection limit. Of course, this is just an 'eyeball' estimate on my part, and may be off by a bit. I suggest the authors do a more careful check and add the details to the text.

Ideally, the gain on one of the SP2's incandescent channels should have been set so as to extend the detection limit. The fact that it wasn't does not necessarily damage the story the paper is telling, in my opinion. . .but nevertheless, more care should be given to estimating the effects of this. Ultimately, the SP2-determined rBC concentrations presented in the manuscript are more appropriately viewed as low-bounds until corrected for the undetected rBC mass. Note also that accounting for the undetected rBC would bring the SP2-determined concentration and the PAAS-3L calculated-concentration into better agreement.

Third, on Line 192: The concentration of the PSL standard is higher than I'm comfortable using for my own Marin-5+SP2 setup. . .I wonder if there was any evidence of multiple PSL particles existing within a single SP2 trigger? This could affect the efficiency calculation. It's easy enough to look through the raw SP2 data and confirm that this isn't a common occurrence for their data. I'm not claiming that I expect that there is a major issue, but simply that it should be looked for when using concentrations that high. Further, the authors do not include the size of the PSLs. . .this should be stated.

Finally, in general, the writing needs some 'smoothing', as certain word choices and sentence structures don't flow as well as they could. I would recommend a careful proofreading.

Technical Corrections:

General comment: I see no need to continually put 'fullerene' in parenthesis

Line 56: extend -> extent

Line 64: I don't like the phrasing "should be" in this circumstance. I'd suggest something like "The authors determined albedo values of only 0.5-0.7 for the ultraviolet and visible range, substantially lower than the 0.97-0.99 that is typical for clean snow [include a citation]"

Line 67: "Most Himalayan glaciers as glaciers. . ." - > Should this be "Most Himalayan glaciers and glaciers" ?

Line 74: smooth wording of sentence beginning "In the last decade. . ."

Line 96: analyses -> analyzes

Line 118: remove word 'used'

Line 207: the authors use the word 'daily' twice in quick succession. Again, smooth the writing here

Line 221: depending -> dependent

Line 225: the phrase "it turned out" is too colloquial for a scientific paper

Line 268: remove the word 'used'

Line 282: please add the word 'these' to specify that this statement isn't generally true (for instance, in the case of snow that has experienced freeze/thaw), i.e. "Thus, the majority of the rBC particles in these fresh snow samples have. . ."

Line 322: The use of the word "therefore" is not appropriate here. Recommend rewording.
* * *
[Figure]

NOTE: Black 'plus' symbols are just my best attempt to extract the Size distribution data from the authors plots might be off a bit, but doesn't affect my point.

**Fig. 1.**

---

## Referee Comment (RC3) · Anonymous Referee #1 · 26 Feb 2019

Review of Linke et al. 'Specifying light absorbing properties of aerosol particles in fresh snow samples, collected at the Environmental Research Station Schneefernerhaus (UFS), Zugspitze.'

**General comments**:

This paper present results from snow samples analyzed for their aerosols content over one winter season. The authors are using an array of different instrumentation, making this an interesting study that is most certainly within the scope of ACP. In light of the instrumentation used in this study, it appears that the authors have not presented the full potential of the data that should be at hand in the current version of the manuscript. For example: absorption measurements coupled with the BC mass content have been carried out, enabling the authors to present direct MAC values for the particulates in the snow (which could then be used for comparison and additional data in the manuscript). Another issue that should be addressed more thoroughly in the manuscript, is the fact that it seems as there is only the one sample (from March 10th) where the additional analyzes are performed, and from this one sample conclusions about all of the snow samples are drawn. Please see further comments below on these topics, as well as some other questions and recommendations that should be addressed. Lastly, the manuscript's language should be checked and re-evaluated for a better read. Although I'm not a native English speaker myself, I do believe the manuscript would benefit from such a procedure. Some mistakes have been highlighted in the technical corrections below, but I'm sure there are mistakes that I have missed, and so make sure to check the whole manuscript. On the whole, this work would be a welcomed addition to the literature after it has been majorly revised.

**Specific comments**:

Section 1

The introduction could use some restructuring and clarifications. My opinion is that some references are missing, and the addition of these papers will change some of the claims the authors are making in the introduction. Under 'technical corrections' I have provided comments to specific line numbers on this.

Section 2

Lines 128-129: Do the authors have any idea to what degree the station is affected by the anthropogenic emissions? It would be valuable information to add in the text, if it exists? How close is UFS to the skiing area?

Line 136: As the manuscript currently reads, 10 cm of snow was collected after each snowfall. What was the procedure of collecting the snow if there was more (or less) snow in the precipitation event? If it was more than 10 cm, for example, BC particles from the beginning of that snow event would be missing in the analysis. On the opposite, if there was a small snow event, producing only a few centimeters of snow, how was that snowfall sampled? In that event, if 10 cm of snow was sampled, it would include not only the fresh snow (and its particulates), but also more aged particles in the snow, leading to possible differences in the analysis.

Also, on the topic of the snow sampling, the authors mention that the collection was from a place exposed to wind, and so how did they manage to collect the new precipitation? One would expect the wind to remove this snow (that is typically is not very dense). A table in the supplement of this manuscript could easily be added, providing valuable information about the snow samples, precipitation events, as well as other basic weather parameters during sampling.

Line 144: How do the authors know that using the ultrasonic bath to melt the snow sample did not change the structure of the BC particles? (e.g. break the particles apart into smaller sizes, which obviously will have an impact on their optical properties), or the other particles present in the snow also?

Section 3

Some sentences in the fourth paragraph are already mentioned in the second paragraph (e.g. lines 235-237 are mentioned in lines 216-219). Technically, I do not see why the information in this fourth paragraph (which ultimately contains more details) could already be incorporated into the second paragraph.

Lines 225-226: Could you present any quantitative numbers on this minor influence?

Section 4

Lines 263-266: What about adding the size distribution of the fullerene in a figure, to show that the difference in fullerene MAC's could be explained by differences in size?

Section 5

The results of the snow sample measurements could be highlighted even more. You have 33 samples from one snow season. How do they differ from one another? In Fig. 7 you show how the BC snow concentration varies over this season, what about the air concentration of BC from the station? (it is mentioned that no correlation was found between the snow and air BC concentrations. I would argue that this is shown, possibly in fig. 7 or a different figure). Is there any seasonality in the size distribution of BC particles from snow? I'm not convinced that there is such a clear decrease in the size distribution of the larger BC particles, please elaborate on this to further convince the reader that this is the case. (It is true that since you have taken relatively fresh snow samples, there would be no thawing and freezing cycles, possibly leading to larger BC particles). One example comes to mind where fresh snow BC size distributions were measured (Sinha et al., https://doi.org/10.1002/2017JD028027). Although in a different setting, please put your results into context by comparing with this study.

What about the dust concentrations? How did the described Saharan dust episodes influence the BC mass (and the snow samples)?

Through the measurement set-up that the authors present, they should have the data necessary to directly derive MAC values for the particles in the snow. How come this was not done? I believe one of the other referees also commented on this. Either the authors provide these MAC values also, and compare (and discuss) that in the manuscript with the data that they already have. If this is not possible, then it should be stated why, and more emphasized why the approach used currently in the manuscript is utilized. There is evidently other impurities other than the BC particles, which will influence the MAC

value for the particles in the snow, but with the other instrumentation available, it should assist in describing those particles (i.e. the OC content).

Line 313: What evidence is there to show that this one sample from March 10th is representative for all of the snow samples? (This sample contains a low amount of BC compared to the other samples and has an enhancement factor of 2.7 compared to the 2.34 presented for the others samples). Either present some evidence that this is representative for all of the samples or emphasize in the manuscript that these additional analyzes (presented in the following paragraphs) were only done to the one sample, and so it is difficult to draw conclusions for all of the snow samples. Ideally, I would argue that additional snow samples would be analyzed from different times during the season in the same way as the one sample from March 10th.

Lines 320-326: Although you discuss it further in the following paragraph, please include some sentence (or sentences) of what these specific results indicate about the snow samples?

Line 329: It would be interesting to have some information (even if it is hypothetical) on where this biological information originate from? E.g. Local or long distance?

**Technical corrections**:

Lines 29-30: This opening sentence is not structured well. Please revise for a better read.

Line 32: Please remove 'packs' from ice packs. Could say snowpack, but not ice packs here.

Line 32-33: You could argue that a better reference here would be Warren and Wiscombe 1980 https://doi.org/10.1175/1520-0469(1980)037<2734:AMFTSA>2.0.CO;2, look into details of Doherty et al. 2010.

Lines 33-34: How is this sentence different than the previous sentence? I would think that it is better to have this sentence earlier.

Lines 35-36: How does 'this reduction' contribute to the snow-albedo feedback? Please include in the manuscript. What metamorphosis?

Line 41: What 'BC amounts' are you referring to? Please specify.

Lines 44-46: How are permafrost regions also affected? Unclear what you mean how they are affected? Also, after reading the rest of this paragraph, I would argue that you should remove this sentence. Since the rest of the paragraph discusses the Arctic, and these other 'areas' are not brought up again until later in the introduction, it could come then instead.

Line 50: As far as I remember Flanner et al. (2007) did not present any measurements, but based their modeling work on measurements instead.

Line 53: I would argue that you either introduce what the term 'soot' refers to, or stick with only discussing BC.

Line 56: Ice sheet instead of 'ice shield.' Please change also 'extend' to extent.

Lines 55-60: Concerning the Greenland ice sheet, there are also new papers on this topic of impurities, which could be added here (e.g. https://doi.org/10.5194/tc-10-477-2016; https://doi.org/10.5194/tc-11-2491-2017).

Line 60: Please capitalize a in 'arctic.'

Line 61: Doherty et al. (2010) is already referenced to in the beginning of the sentence.

Line 68: How high amounts of dust? Would be more informative to actually reference to some numbers on this.

Line 69: To my knowledge, Bolch et al. (2012), is incorrectly referenced to here. No studies of BC nor dust were conducted in that paper.

Line 73: Remove 'the' before light absorbing particles.

Lines 77-80: I find this sentence confusing, please rewrite. Mixing the optical method (Doherty et al., 2010), and then the thermos-optical analysis, with the previous sentence about MAC causes some of the confusion. The following paragraph (lines 81-93) dig deeper in each analysis technique and that is appropriate, but the order of this seems strange, in light of the previous paragraph. Actually, I think you could delete the sentences in lines 75-80, and jump right into line 81 and an explanation of the methods (after current sentence ending on line 75).

Lines 92-93: Either remove this sentence or add more information on other instruments and protocols out there (e.g. DRI and Improve-protocol). I would vote for removing this sentence, I do not think it is very crucial information.

Line 97: I would argue that you do not need quotation marks around Fullerene.

Lines 98-100: This sentence is basically a repeat of the first sentence of this paragraph, please remove.

Lines 102-103: I generally agree with this statement that not much have been reported on the light absorbing properties. But, there has been some publications on this topic addressing it directly and indirectly, e.g. Schwarz et al., 2013 https://doi.org/10.1038/srep01356; Zhang et al., 2017 http://dx.doi.org/10.1016/j.scitotenv.2017.07.100; Dal Farra et al., 2018 doi: 10.1017/jog.2018.29; Dong et al., https://doi.org/10.5194/tc-12-3877-2018. Please add and discuss these references.

Line 104: Please remove 'solar' before albedo.

Lines 103-106: Please clarify the structure by checking the structure. As it currently stands, it is not clear what the main point of the sentence is.

Lines 106-108: Please change this sentence according to the forthcoming changes made for lines 102-103.

Lines 109-110: The second half of this sentence (starting after 'but) I find problematic. Similar to the comment in lines 102-103, I do believe this topic has been addressed in the literature. For example: Kaspari et al., 2014 (that you already referenced to earlier in the manuscript); Skiles and Painter, 2016 doi: 10.1017/jog.2016.125; Schmale et al., 2017 DOI: 10.1038/srep40501; Zhang et al., 2018 https://doi.org/10.5194/tc-12-413-2018, 2018. Please adjust your claim by including these references on this topic.

Line 137: The fact that the snow samples were collected at 'platform 7' does not add any information to a reader unfamiliar with UFS. Please either elaborate on this, or remove.

Line 147: What does 'Enhanced' refer to? Please explain.

Line 150: I do not find the flow rate for this peristaltic pump anywhere. Please add it.

Line 184: Please remove the double reference to 'Fischer and Smith (2018).'

Lines 196-198: Do you mean that the solution was prepared in the same way as in Schwarz et al? If so, please correct. Also, I believe the reference should be Schwarz et al. (2012) and not (2010) as it currently reads.

Line 204: How did you 'drop' 30 mL of fullerene solution onto the quartz filters? Please explain more.

Line 260: What did Zhou et al. (2017) refer to with $MAC_{real}$? Please clarify.

Line 276: Remove 'before' at the end of the sentence.

Line 294: The presented enhancement factors for the different wavelength appear to be averages, please clarify this.

Lines 315-319: The instruments and methodology presented here should be described in section 2.

Lines 344-345: This information should be moved to section 2.

Figure 2. I'm not sure how needed this figure is. I actually think that this figure could be integrated into fig. 1.

Figure 8. Why is there is a data gap around 280-300 nm?

Figure 9. This figure is quite busy right now. Could the data points be zoomed in on more? And could the data points be made smaller?

---

## Author Comment (AC1) · 16 Jun 2019

We thank the anonymous reviewer for the helpful comments. These comments helped to substantially improve the manuscript. Below we give detailed answers to the individual reviewer comments in blue.

Manuscript: Specifying light absorbing properties of aerosol particles in fresh snow samples, collected at the Environmental Research Station Schneefernerhaus (UFS), Zugspitze (Linke et al.,)

It is often regarded that the polar regions are the "canary in the coal mine" regarding this region's sensitivity to changes in radiative forcing. This sensitivity derives, in part, from the very high surface albedos that typify these regions. However, despite the recognized importance of this subject, our quantitative understanding of the radiative contribution by light absorbing aerosols - be it atmospheric or surface-deposited particles - in these regions - and on snow/ice - is still very low. Therefore, more research is indeed needed to improve our understanding of this subject and to reduce the associated uncertainty. To this end, the manuscript by Linke and co-workers undertake an investigation of light absorbing properties of aerosols deposited on snow near the Environmental Research Station Schneefernerhaus with the goal to better quantify the aerosol types (classes) of light absorbing aerosols present in this region. To derive the optical properties, these researchers combine aerosol light absorption measured by a photoacoustic spectrometer with refractory black carbon (rBC) mass loading measurements reported by a single particle soot photometer (SP2) on re-aerosolized snow samples. The insights and data that can be gleaned from a study such as this is of value and should, eventually, be published. But in its current state, the manuscript is not ready and thus is not recommended for publication at this time.

One of the biggest issues this reviewer has centers on how these researchers are combining their light absorption and rBC mass measurements. In the atmospheric aerosol community, we take these two datasets and derive the mass absorption cross- section (MAC; m2/g). This property enables researchers to say something about the mixing state of the rBC particles as well as something about aerosol type (e.g., the presence/absence of light absorbing organic material/particles (e.g., brown carbon or BrC and dust). However, despite the ubiquity of this methodology in our community, the authors, instead, advocate using a methodology that compares the SP2-derived mass with what they term as a "fullerene soot equivalent mass" derived from the photo acoustic spectrometer. In order to use a "fullerene equivalent mass" derived from the light absorption measurements the researchers must assume that the MAC is constant - the literature is populated with many studies that show that the MAC is not constant. For example, at 550 nm, the black carbon MAC for fresh (uncoated) soot is ∼ 7.5 m2/g while for thickly-coated black carbon particles this value could be 12-14 m2/g at this wavelength, thus representing a factor of 2x change. How do the authors account for the potential changes in the MAC brought about by changes in the rBC mixing state? Why not simply derive a MAC and utilize the variability in this value to infer something about the light absorbing particles deposited on the snow? The authors provide no justification as to why their methodology is preferred. If the authors feel strongly that this methodology offers an advantage not available with a MAC-based methodology, then they need to present that argument. Further, any sources of error in the interpretation of the data using this methodology is not addressed and needs to be.

The referee can be sure that the authors know the importance and the ubiquity of the MAC in the atmospheric aerosol community as they published important contribution to this field in the past (e.g. Schnaiter et al., 2003; Schnaiter et al., 2005; Schnaiter et al., 2006). The authors used the optical equivalent BC mass here because it was used in one of the most comprehensive studies on light absorbing impurities in Arctic snow by Doherty et al. (2010). Also, the authors are aware of the fact that the MAC is not constant in the atmosphere and the MAC of thickly coated soot can be enhanced by a factor of two, as they provided the first experimental proof of this effect already in 2005 (Schnaiter et al., 2005). However, the authors agree that the presentation of the absorption measurements is insufficient in the

manuscript. Therefore, we did a thorough reanalysis of the snow samples following the following strategy that also includes a MAC analysis of the data:

1. The snow mass specific absorption cross section $\sigma_{abs}$ [m$^2$/mL] was deduced from the absorption coefficient $b_{abs}$ [m$^{-1}$] using the nebulizer flow settings and the nebulizing efficiency (Eq. 1 of the revised manuscript):

$$\sigma_{abs} = 10^{-3} \cdot b_{abs} \cdot R_{neb}/R_{pp} \cdot \varepsilon_{neb}^{-1}$$

2. This snow mass specific absorption cross section is plotted as a function of the refractory BC mass concentrations in a new figure (Fig. 10 of the revised manuscript). Linear regression fits to the data then give the BC mass specific MAC of the snow samples. We found MAC values that are up to a factor of two larger than the MAC of Fullerene soot. A table is added to the manuscript contrasting the mass and optical properties (including the MAC) of Fullerene soot with those of the snow samples.

3. To be comparable with other studies (e.g. Doherty et al., 2010), we calculated the equivalent BC mass concentration $c_{BC}^{equiv}$, i.e. the amount of BC that would need to be present in the snow to account for the measured absorption, from $\sigma_{abs}$ using the MAC of Fullerene soot (Equation 2 of the revised manuscript):

$$c_{BC}^{equiv} = \sigma_{abs}/\text{MAC}_{FS}$$

4. We plotted $c_{BC}^{equiv}$ as a function of the refractory BC mass concentration in a new figure (Fig. 11 of the revised manuscript) and found a good correlation of both concentrations but with correlation coefficients of 2.0, 1.9, and 1.4 for 405, 532, and 658 nm, respectively (note that $c_{BC}^{equiv}$ is a function of the wavelength). We conclude from this that there is additional non-BC light absorbing mass in the snow, which is correlated with the BC mass and which has a strong wavelength dependence between the green and the red part of the visible spectrum. This already indicates mineral dust and organic (brown) carbon as possible carriers of this additional absorption.

5. We calculated the snow mass specific absorption cross section of the non-BC particles, $\sigma_{abs}^{nonBC}$ (Equation 3 in the revised manuscript):

$$\sigma_{abs}^{nonBC} = \sigma_{abs} - c_{BC}^{SP2} \cdot \text{MAC}_{FS} \cdot 10^{-9}$$

We added a figure (Fig. 12 in the revised manuscript) that shows the statistical analysis of the $\sigma_{abs}^{nonBC}$ for the snow samples and that compares this non-BC spectral behavior with laboratory data for Saharan dust and organic (brown) carbon (see answer to the previous comment).

Continuing, this reviewer is very concerned about the reported MAC values for the fullerene standard. Fullerene soot should be uncoated and thus should exhibit a MAC that is characteristic of an uncoated black carbon particle, namely a MAC ~ 7.5 m2/g at 550 nm. Yet, at 532 nm, the authors report a MAC of of 10.6 (+/- 2.8) m2/g - a value that is ~40% that the canonical value for denuded soot, and 65% larger than the fullerene MAC reported by Zhou (6.4 m2/g) - work cited by the authors. Additionally, it is also at odds with the recently published value by Zangmeister 6.1 m2/g at 550 nm (Carbon (2018), doi: 10.1016/j.carbon.2018.04.057). The disagreement is quite critical as it will have bearing on interpretation of data collected on the snow samples. While the authors are correct that the BC MAC is dependent on particle size, using the argument that the differences between their higher values and that reported by others

as being due to differences in the sample size distribution studied is not correct. Using a refractive index of 1.95-0.79i (Bond & Bergstrom (2006); Aero. Sci. Tech. 40:1, 27- 67) a straight forward Mie calculation reveals that the maximum MAC calculated for a BC is ~7.9 m2/g for a BC diameter of 150 nm and that the MAC only goes down with either increasing or decreasing particle size. Therefore another explanation is needed. Three potential explanations submitted here are: (i) either the fullerene soot is coated and the derived MACs reflect a lensing-enhanced value or (ii) there are some non- BC, light absorbing aerosols present in the sample, or (iii) the number concentrations used during the fullerene soot calibration are high enough that particle coincidence is occurring that the post-processing of the SP2 data does not correct for. This latter possibility is brought up because particle-resolved measurements, like the SP2, can easily find themselves in the particle-coincidence regime. Such an undercounting of the actual fullerene mass would bring the reported MACs down and thus reconcile their MAC values with others. The authors are encouraged to insure that the SP2 is not operating in this regime.

We do not agree with the Referee that the MAC of uncoated BC is canonical with a constant value of 7.5 m2/g. Only because something is continuously referenced does not necessarily mean that it is correct. A constant MAC would mean that refractive index of the material is constant. From a solid-state physical perspective this is highly uncertain in case of carbonaceous material with its variable electronic band structures. Already pure carbon can exist in five different allotropes including graphite and diamond – one is black the other transparent – so obviously completely different refractive indices. Adding now the morphological variability of fractal soot particles to the discussion with a wealth of different monomer sizes, monomer nonsphericity (irregularity), necking, and overlapping (even on a single particle) that all have an influence on the particle absorption cross section, it is conclusive to the authors that the absorption cross section of this material cannot be constant but depends on its formation conditions. Further, the MAC is calculated from the absorption cross section and the particle mass. Particle mass is measured in different ways – sometimes as a total mass (including non-refractory compounds as in the DMA-APM method), sometimes as a refractory mass only (as with the SP2). So, there comes an uncertainty also from this side into the MAC. It is correct that Fullerene soot was identified as a good standard for atmospheric BC in terms of the SP2 mass sensitivity (although with a variation of 15% from batch to batch). But does that mean that this is also the case for its spectral absorption properties given the variability in the electronic and microphysical structures discussed above? With all this, the authors do not see any reason why a laboratory to laboratory difference in published MAC values of 30-40% shouldn't be within the variability range of the material itself, the size differences and the mass measurement methods used in the different studies.

The authors took care that the SP2 wasn't operated under aerosol concentrations that would result in coincidence errors. However, it turned out that due to the upper cut size of the SP2 (560 nm), we missed about 10% of the Fullerene particle mass. In the reanalysis we therefore applied lognormal fits to the measured BC mass size distributions to account for this mass outside the SP2 sizing range (see Fig. 6 of the revised manuscript). With this correction the MAC of FS is reduced to 9.5 ± 2.2 m2/g (532 nm). Further, to be comparable with the Zangmeister et al. (2018) study who measured the MAC on size-selected FS particles, we also measured the MAC of size selected Fullerene soot particles in a separate study by adding a DMA behind the Marin-5. We found a MAC of 8.6 m2/g for the same mobility-equivalent diameter (350 nm) as used in the Zangmeister et al. study. This is still 40% larger than the 6.1 m2/g of Zangmeister et al., but given the fact that (i) a APM/DMA was used for the mass measurement in their study and (ii) they used a different batch of Fullerene soot, this difference is within the range of variability that can be expected in such a comparison.

The whole Sect. 4 was rewritten to make the discussion of the Fullerene soot MAC clearer:

"Simultaneously to the BC mass concentration measurements with the SP2, the absorption coefficients $b_{abs}$ of the Fullerene soot suspensions were measured for the three PAAS-3λ

wavelengths. Both measurements together enable the determination of the mass specific absorption cross section $MAC_{FS} = b_{abs}/c_{BC}^{SP2}$ of airborne Fullerene soot at 405, 532, and 658 nm. In **Error! Reference source not found.**, the absorption coefficients $b_{abs}$ are plotted against the SP2-derived BC mass concentrations $c_{BC}^{SP2}$ of the Fullerene soot suspension standards. Linear regression fits of the data result in $MAC_{FS}$ values of 10.5 ± 3.2 m2/g, 9.5 ± 2.2 m2/g, and 8.6 ± 3.3 m2/g for 405, 532, and 658 nm, respectively. The $MAC_{FS}$ at 532 nm is comparable to the value of 8.84 m2/g given by Schwarz et al. (2012) for Fullerene soot (lot #F12S011) deduced from ISSW measurements, but is significantly higher than the 6.1 ± 0.4 m2/g (mean ± 2σ) measured recently by photoacoustic absorption spectroscopy for size selected Fullerene soot particles by Zangmeister et al. (2018). They used a combination of a differential mobility analyzer (DMA) and an aerosol particle mass analyzer (APM) to select Fullerene soot particles within a narrow mass range from aerosol generated by an atomizer. Their $MAC_{FS}$ of 6.1 m2/g, which is given for a wavelength of 550 nm, a mobility-equivalent diameter of 350 nm, and for a particle mass of $16.6 \cdot 10^{-15}$ g, corresponds to a volume-equivalent diameter of 264 nm using a density of 1.72 g/cm3 of Fullerene soot (Kondo et al., 2011). Although, this diameter is not very different to the MMD of 228 nm of the Fullerene soot suspensions used here, part of the observed discrepancy can be attributed to the different sizes as the MAC is strongly depending on the particle diameter for particles larger than about 200 nm (e.g. Moosmüller et al., 2009). To be comparable, we measured the $MAC_{FS}$ of size selected Fullerene soot particles in a separate study by adding a DMA behind the Marin-5 in the setup shown in **Error! Reference source not found.**.  A $MAC_{FS}$ of 8.6 m2/g was measured for the mobility-equivalent diameter of 350 nm, which is still ~40% larger than the $MAC_{FS}$ given by Zangmeister et al. (2018) for the same diameter. However, they used an APM to measure the BC mass, while a SP2 was used here to deduce the refractory BC mass. According to Laborde et al. (2012a), the Fullerene soot product shows a variability from batch to batch, which results in a SP2 calibration uncertainty of up to 15% (actually only two batches were compared; lot #F12S011 and lot #L18U002). They explained the differences in the SP2 response (i.e. the calibration curves) by a substantial non-refractory coating in case of the L18U002 batch that could be identified by thermodenuding the samples. Assuming that lot #W08A039 used in Zangmeister et al. (2018) has a similar coating, this would increase the APM mass measurement by about 15% compared to the SP2-derived BC mass of lot # F12S011 used in the present study. This in turn would increase the $MAC_{FS}$ from 6.1 m2/g reported by Zangmeister et al. (2018) to about 7 m2/g when using only the refractory BC mass fraction in the calculation of the $MAC_{FS}$. This assumption reduces the discrepancy between the two $MAC_{FS}$ values to 35%, which is within the uncertainty range of ± 2.2 m2/g for our 532 nm value. It is further conceivable that different batches of the Fullerene soot material have different electronic band structure (i.e. refractive index) and/or fractal aggregate structures that both change the absorption cross section of the particles at a constant particle mass (e.g. Liu et al., 2019; Zangmeister et al., 2018). **Error! Reference source not found.** shows an electron micrograph of a typical Fullerene soot aggregate sampled from the dry aerosol output of the Marin-5 nebulizer. Thus, the Fullerene soot particles do not have a simple fractal aggregate structure, but are rather complex-structured with polydisperse monomer sizes, monomer nonsphericity (irregularity), necking, and overlapping, which all have a significant impact on the optical particle properties (including the absorption cross section) compared to the idealized fractal aggregate (Teng et al., 2019). Since these microphysical details of the soot particles are very sensitive to the actual formation and subsequent treatment conditions (Gorelik et al., 2002), it is conclusive that the $MAC_{FS}$ has an even higher variability between different Fullerene soot batches compared to what is expected from the SP2 mass sensitivity only.

The wavelength dependence of the aerosol light absorption, expressed by the so-called absorption Angström exponent (AAE), was determined to be 0.46 ± 0.07 for the used Fullerene soot suspensions by analysing the $b_{abs}$ data for the 405 and 658 nm wavelengths. This AAE is close to the ~0.6 reported by Baumgardner et al. (2012) for Fullerene soot derived from multiwavelength PSAP and Aethalometer measurements and it is within the range of the 0.54 ± 0.06 determined by Zhou et al. (2017) from ISSW spectrometer

measurements on Fullerene soot filter samples in the 450 nm to 750 nm spectral range. However, it is significantly lower than the 0.92 ± 0.05 given by Zangmeister et al. (2018) for Fullerene soot lot #W08A039. Here again, we have to take into account that Zangmeister et al. (2018) analyzed size segregated absorption spectra and their AAE is given for a mobility-equivalent diameter of 350 nm. Analyzing our size segregated measurements gives an AAE of 0.82 ± 0.02 for the same mobility-equivalent diameter, which is close but smaller than the Zangmeister et al. value, further supporting the above assumption that there is a difference in the chemical as well as physical (including optical) properties between different batches of the Fullerene soot product."

While on this subject, what were the re-aerosolized number concentrations in the snow samples studied? On lines 243-244, the authors state that ". . ..Marin-5 nebulizer was then fixed at a rate of 0.32 mL min-1, which guarantees a high enough particle mass concentration for the photo acoustic measurement. Depending upon the mass concentration and mode size of the particles, this could hint at co-incidence issues with the SP2.

The concentrations behind the Marin-5 were always low enough (< 1000 #/cc) to avoid any coincidence issues.

Other specific issues:

Do the authors worry about the loss of water soluble BrC during sample preparation that might not be reflected when re-aerosolizing their snow samples?

We do not have any information on potential soluble BrC mass loss.

Line 167: What were the PSL diameters used? While the incandescence channel of the SP2 is sensitive enough to detect <100 nm rBC particles, the scattering channel is typically limited to optical diameters > 200 nm. What is the particle number loading in the instruments (see reference to coincidence above)

The characterisation of the particle number efficiency of the nebulizer and the daily performance control was performed with monodisperse polystyrene latex (PSL) particles (Postnova Analytics GmbH, Landsberg am Lech, Germany) with nominal diameters of 240 ± 5 nm and 304 ± 5 nm. The particle number concentration within these suspensions is about $3 \cdot 10^8$ 1/mL. A diluted PSL standard suspension sample was prepared daily by pipetting 1 mL suspension into a 100 mL graduated flask filled with Nanopure water. This results in a concentration of typically a few hundred particles per cc at the output of the Marin-5.

Line 291-293: The authors report and discuss the "enhancement" factor of the soot- equivalent mass derived from the photoacoustic spectrometer with decreasing wave- length (1.65, 2.28, and 2.38 at 658 nm, 532, 405 nm, respectively) and conclude that this suggests these samples might contain mineral dust or BrC. The data might be able to say something more concrete. While the authors should conduct a more thorough literature search, mineral dust tends to absorb more in the red then the blue, whereas BrC exhibits the opposite wavelength dependence. The wavelength dependence of light absorption observed in the present study suggests that the non-rBC absorbing species is BrC. Also, it should be noted that BrC can exhibit absorption at the red wavelengths, though, again, the wavelength dependence favors more absorption in the blue (e.g., tar balls: Hoffer, et al., 2017. Brown carbon absorption in the red and near infrared spectral region. Atmos. Meas. Tech. 10(6): 2353–59).

Although we cannot directly measure the dust concentrations in the snow samples, we can draw some conclusion by analyzing the spectral signature of the snow mass specific absorption cross section of the non-BC particles, i.e. after subtraction the spectral cross section that is expected from the SP2 mass data using the MAC of Fullerene soot (Equation

3 of the revised manuscript; see Point 5 above). We added a figure (Fig. 12 in the revised manuscript) that shows a comparison of the non-BC spectral absorption cross section with laboratory absorption data for Saharan dust particles. Thus, Saharan dust is a good candidate to explain the observed non-BC light absorption in the snow samples. We did the same comparison with laboratory data of OC, which reveals that OC could also explain the observed non-BC absorption - not only in terms of wavelength dependence but also in terms of variability.

The whole paragraph on the analysis of the spectral absorption analysis has been rewritten to make our argumentation of a significant influence of non-BC particles on the light absorption in snow clearer.

line 349: "The term brown carbon is not clearly defined or characterized." This is very misleading. Simply put, brown carbon, BrC, is organic aerosol that absorbs light. While the chemical composition for any given BrC aerosol may very, as long as the organic aerosol absorbs light it is cataloged as BrC. Please reword this sentence.

Paragraph rephrased to:

"The term "brown carbon" is mainly related to a strong wavelength dependence of the visible light absorption observed in these materials. From a chemical perspective, BrC can generally be divided into humic-like substances (HULIS) and tar balls (Wu et al., 2016). HULIS can be characterised mainly as a mixture of macromolecular organic compounds with various functional groups and are expected e.g. in oxidation processes of biogenic precursors (Wu et al., 2016). Tar balls are emitted from biomass burning and are of spherical, amorphous structure and are typically not aggregated. Moreover, light absorbing organic material and HULIS can be formed from the water-soluble fraction of biomass burning aerosol compounds, and is therefore suggested as an atmospheric process for the formation of light absorbing BrC in cloud droplets (Hoffer et al., 2004). Further examination of snow samples from different locations as well as systematic investigations on the optical behaviour of biogenic particulate matter is therefore necessary to evaluate the influence of biogenic (including biological), BrC and mineral dust on the aerosol absorption properties in the visible spectral range."

Please provide plot of fullerene standard size distribution.

Added (Fig. 6 of the revised paper).

Editorial comments

There is no reason to put quotes around the word fullerene.

Agreed and changed.

Line 67: "Most Himalayan glaciers as glaciers elsewhere have retreated..." Please insert comma after first occurrence of the word "glaciers" and after "elsewhere".

Introduction has been completely reworded.

Line 311: "To get a general idea of the nature of the components that are solved and dissolved within the snow samples ionic chromatography . . ." the word "solved" does not seem right here. Please check and correct as necessary.

Reworded to:

"To further examine the nature of the particulate components that are deposited in the snow samples ion chromatography (IC) … "

---

## Author Comment (AC2) · 16 Jun 2019

We thank the anonymous reviewer for the helpful comments. These comments helped to substantially improve the manuscript. Below we give detailed answers to the individual reviewer comments in blue.

Referee Comments for: "Specifying light absorbing properties of aerosol particles in fresh snow samples, collected at the Environmental Research Station Schneefernerhaus (UFS), Zugspitze"

General Comments:

This manuscript presents a novel method for concurrently measuring the rBC mass concentration and total aerosol absorption in snow. The authors find a discrepancy between the SP2-measured rBC concentration and a calculation of rBC concentration that is based on total aerosol absorption and a fullerene-standard's MAC. The former being smaller than the latter, the authors conclude the likely presence of non-rBC absorbing aerosols in the snow. WIBS measurements indicate that the larger of these non-rBC aerosol are predominantly of biological origin. These results are both interesting and important, as the presence and properties of light-absorbing aerosols in snow and their impact on snow/ice melt is not presently well understood.

Overall, the methodology of the measurements seems sound, and the text contains all of the necessary information to understand the experiment. I have four main concerns, three of which are science-related, and one regarding the writing. These can be found in the 'Specific Comments' below. I believe that once these things are appropriately addressed, the manuscript deserves publication in ACP.

Specific Comments:

My first concern is that this study's calculated MAC for fullerene soot is quite high compared to other literature, which typically put it around 7 m2/g or so. The authors acknowledge this, but I don't feel they explore (at least in the text) the underlying cause sufficiently. As this is at least somewhat dependent on the size of the fullerene particles, I suggest that the fullerene standard's size distributions be added to Figure 8.

First, we added a figure (Fig. 6 in the revised manuscript) that shows the refractory BC mass size distributions of our Fullerene suspension standards. Second, these size distributions were fitted by lognormal distributions to get the missed mass beyond the upper size limit of our SP2. The analysis of the Fullerene MAC was than redone with the corrected BC mass, resulting in lower MAC values of 10.5 ± 3.2, 9.5 ± 2.2, and 8.6 ± 3.3 m2/g for 405, 532, and 658 nm, respectively. Third, we completely rephrased Sect. 4 in the revised manuscript that now includes a comparison with published MAC of Fullerene soot and a discussion of potential reasons why our values are larger. This discussion includes the influence of different particle size distributions, different methods to deduce the particle mass, differences in using different Fullerene soot batches as well as the in general higher sensitivity of light absorption to BC electronic band structures and fractal aggregate morphologies compared to the incandescence mass detection.

Second, the authors don't estimate the percentage of rBC in snow that is above the detection limit of the SP2. Again, the authors acknowledge the issue, beginning on line 278, by stating that the SP2 only detects rBC up to 500 nm VED, but then rather simply declare the mass beyond this as small. The size distribution in Figure 8 indicates that there is still non-negligible mass above 500nm, and exactly how much can easily be estimated using a lognormal fit to the distributions. I attempted to estimate this by extracting data from the plot (see my attached figure), doing my own fit, and calculating percentage of area under the curve that is above 500nm. . .I get about 10%. Furthermore, its highly doubtful that the SP2 is detecting with 100% efficiency below 60-70nm or so. This probably has a smaller effect on the total rBC mass, but including that data in a lognormal fit could skew the fit result. If I do the same fit, but only use the size distribution data between 70 – 500 nm, I come up with ~13% of the mass above the detection limit. Of course, this is just an 'eyeball' estimate on my part, and may be off by a bit. I suggest the authors do a more careful check and add the details to the text.

As suggested by the reviewer, we performed a more careful analysis of the refractory BC size distributions of the snow samples. First, we divided the samples in the periods Nov-Jan, Feb-Mar, and Apr-May to check any seasonality in the data. Second, we performed lognormal fits to the data between 70 and 500 nm as suggested by the reviewer and found a mass fraction of 10 to 20% that is beyond the upper size limit of our SP2. Third, we compared our size distributions with the average SP2 size distribution measured for five snow samples collected after three snowfall events in the semi-rural and rural surroundings of Denver, CO, USA by Schwarz et al. (2013). Our size distributions agree very well with the Schwarz et al. distribution, who measured the distribution up to a size of 2 $\mu m$ (see Fig. 9 of the revised manuscript). Due to this good agreement and the fact that the size distributions are skewed with a shoulder towards larger sizes, we decided to use the 28% given by Schwarz et al. for the mass fraction > 600 nm to correct our snow sample SP2 data.

Ideally, the gain on one of the SP2's incandescent channels should have been set so as to extend the detection limit. The fact that it wasn't does not necessarily damage the story the paper is telling, in my opinion. . .but nevertheless, more care should be given to estimating the effects of this. Ultimately, the SP2-determined rBC concentrations presented in the manuscript are more appropriately viewed as low-bounds until corrected for the undetected rBC mass. Note also that accounting for the undetected rBC would bring the SP2-determined concentration and the PAAS-3L calculated- concentration into better agreement.

See answer to the previous comment. With a correction factor of 1.39 that we applied to the SP2 data in the reanalysis (i.e. accounting for a fraction of 28% missed mass), we are rather on the upper limit for this correction. In this way, however, we are confident that the main conclusion of our paper namely that there is a significant contribution of non-BC particles to light absorption in fresh snow is solid.

Third, on Line 192: The concentration of the PSL standard is higher than I'm comfortable using for my own Marin-5+SP2 setup. . .I wonder if there was any evidence of multiple PSL particles existing within a single SP2 trigger? This could affect the efficiency calculation. It's easy enough to look through the raw SP2 data and confirm that this isn't a common occurrence for their data. I'm not claiming that I expect that there is a major issue, but simply that it should be looked for when using concentrations that high. Further, the authors do not include the size of the PSLs. . .this should be stated.

We think that our PSL concentrations are no too high. The manufacturer suspension has a concentration of 3 $10^8$ #/mL, which is further diluted by a factor of 100 before it is used in the characterization of the Marin-5. With the Marin-5 flow settings and the known dispersion efficiency of 36% we end up with a few hundred particles per cc at the Marin-5 output, which is definitely not an issue for the SP2 measurement in terms of coincidence.

Finally, in general, the writing needs some 'smoothing', as certain word choices and sentence structures don't flow as well as they could. I would recommend a careful proofreading.

The manuscript has been thoroughly restructured and reworded.

The technical corrections listed below are addressed in the revised manuscript.

Technical Corrections:

General comment: I see no need to continually put 'fullerene' in parenthesis Line 56: extend -> extent

Line 64: I don't like the phrasing "should be" in this circumstance. I'd suggest something like "The authors determined albedo values of only 0.5-0.7 for the ultraviolet and visible range, substantially lower than the 0.97-0.99 that is typical for clean snow [include a citation]"

Line 67: "Most Himalayan glaciers as glaciers. . ." - > Should this be "Most Himalayan glaciers and glaciers"?

Line 74: smooth wording of sentence beginning "In the last decade. . ."

Line 96: analyses -> analyzes

Line 118: remove word 'used'

Line 207: the authors use the word 'daily' twice in quick succession. Again, smooth the writing here

Line 221: depending -> dependent

Line 225: the phrase "it turned out" is too colloquial for a scientific paper

Line 268: remove the word 'used'

Line 282: please add the word 'these' to specify that this statement isn't generally true (for instance, in the case of snow that has experienced freeze/thaw), i.e. "Thus, the majority of the rBC particles in these fresh snow samples have. . ."

Line 322: The use of the word "therefore" is not appropriate here. Recommend reword- ing.

---

## Author Comment (AC3) · 16 Jun 2019

We thank the anonymous reviewer for the helpful comments. These comments helped to substantially improve the manuscript. Below we give detailed answers to the individual reviewer comments in blue.

Review of Linke et al. 'Specifying light absorbing properties of aerosol particles in fresh snow samples, collected at the Environmental Research Station Schneefernerhaus (UFS), Zugspitze.'

**General comments:**

This paper present results from snow samples analyzed for their aerosols content over one winter season. The authors are using an array of different instrumentation, making this an interesting study that is most certainly within the scope of ACP. In light of the instrumentation used in this study, it appears that the authors have not presented the full potential of the data that should be at hand in the current version of the manuscript. For example: absorption measurements coupled with the BC mass content have been carried out, enabling the authors to present direct MAC values for the particulates in the snow (which could then be used for comparison and additional data in the manuscript). Another issue that should be addressed more thoroughly in the manuscript, is the fact that it seems as there is only the one sample (from March 10th) where the additional analyzes are performed, and from this one sample conclusions about all of the snow samples are drawn. Please see further comments below on these topics, as well as some other questions and recommendations that should be addressed. Lastly, the manuscript's language should be checked and re-evaluated for a better read. Although I'm not a native English speaker myself, I do believe the manuscript would benefit from such a procedure. Some mistakes have been highlighted in the technical corrections below, but I'm sure there are mistakes that I have missed, and so make sure to check the whole manuscript. On the whole, this work would be a welcomed addition to the literature after it has been majorly revised.

We agree with the reviewer that we have not presented the full potential of our measurements and data. That's why we have made efforts to include a more thorough analysis of the data, resulting in additional discussions (which includes now also comparisons with literature data) of (a) the MAC of "Fullerene" soot, (b) the MAC of the snow samples, and (c) the spectral absorption of the non-BC particles in the snow. While the ESEM and fluorescence analysis of further snow samples is certainly something that would strengthen our conclusions, we refrained from doing them simply because these analyses (especially with the ESEM) are very time consuming and expensive, which we couldn't invest at this time for this project. Finally, the manuscript has been improved in language.

**Specific comments:**

Section 1

The introduction could use some restructuring and clarifications. My opinion is that some references are missing, and the addition of these papers will change some of the claims the authors are making in the introduction. Under 'technical corrections' I have provided comments to specific line numbers on this.

The introduction was completely restructured rewritten taking into account the comments provided in 'technical corrections'

Section 2

Lines 128-129: Do the authors have any idea to what degree the station is affected by the anthropogenic emissions? It would be valuable information to add in the text, if it exists? How close is UFS to the skiing area?

Actually, there exists a study on the influence of anthropogenic activities on the UFS station by Yuan et al. (2019). They detected on weekdays multiple short-term atmospheric CO events and higher atmospheric NO peaks during the daytime (mostly around 09:00 LT), which were interpreted to originate by anthropogenic working activities and less by tourists. During the period the snow samples were collected, the 1 min ambient air eBC data show elevated levels above 1 µg/m3 only on 20 days and only for short periods of less than 20 min giving a total time of about 160 minutes the station experienced this higher BC pollution levels. This time is less than 1 permille of the total snow sampling period. For those days we have no indications that the snow samples are affected by anthropogenic emissions. However, the results of two samples collected on February 6 and February 17 were discarded from the data, because they show inexplicable high BC mass mixing ratios and absorption coefficients (factor 5 to 10 outside the 95th percentile of the other samples), which indicates a possible contamination from local sources.

Line 136: As the manuscript currently reads, 10 cm of snow was collected after each snowfall. What was the procedure of collecting the snow if there was more (or less) snow in the precipitation event? If it was more than 10 cm, for example, BC particles from the beginning of that snow event would be missing in the analysis. On the opposite, if there was a small snow event, producing only a few centimeters of snow, how was that snowfall sampled? In that event, if 10 cm of snow was sampled, it would include not only the fresh snow (and its particulates), but also more aged particles in the snow, leading to possible differences in the analysis.

We made the procedure clearer by rephrasing the paragraph in Sect. 2.1 as following:
"The snow samples were taken either during or just after snowfall events by scraping off only the top few centimeters of the snowpack to avoid sampling older snow. A metallic hand shovel is used to sample the snow from an area of about 30 x 30 cm into a zipper sealed polyethylene household plastic bag with a volume of 1 L (Toppits, Germany). In this way, snow from the beginning of the snowfall event could be missed, but most of the time the events were accompanied by heavy wind, so that it was impossible to completely sample the fresh snow layer. After collection, the samples were stored at the UFS in a freezer at -18°C until they were transported under frozen conditions to the laboratory at KIT. During six months, 33 samples were taken at the UFS."

Also, on the topic of the snow sampling, the authors mention that the collection was from a place exposed to wind, and so how did they manage to collect the new precipitation? One would expect the wind to remove this snow (that is typically is not very dense). A table in the supplement of this manuscript could easily be added, providing valuable information about the snow samples, precipitation events, as well as other basic weather parameters during sampling.

In addition to the changes in Sect. 2.1 given above, we replaced Figure 7 with a stacked figure (Figure 8 in the revised manuscript) that highlights the trends in ambient temperature, sunshine duration, snow precipitation and snow height over the period the snow samples were collected at the UFS station. In this Figure also the ambient eBC and the refractory BC concentration of the snow samples is shown to give the reader an overview about the ambient conditions and their changes during the 6 months period of snow sampling.

Line 144: How do the authors know that using the ultrasonic bath to melt the snow sample did not change the structure of the BC particles? (e.g. break the particles apart into smaller sizes, which obviously will have an impact on their optical properties), or the other particles present in the snow also?

We have no information how the ultrasonic bath might affect the microstructure of the particles. However, we followed the recommendations that are given in the literature concerning snow sample analysis to be at least consistent with those studies. We added the following paragraph to the manuscript to make this clearer:

"Aqueous snow/ice sample sonication prior to the analysis is recommended by several groups (e.g. Kaspari et al., 2011; Wendl et al., 2014) although with inconclusive results of the obtained improvements. The melted samples were never refrozen for later analysis as this can result in a significant particle mass loss of up to 60% (Wendl et al., 2014)."

Section 3

Some sentences in the fourth paragraph are already mentioned in the second paragraph (e.g. lines 235- 237 are mentioned in lines 216-219). Technically, I do not see why the information in this fourth paragraph (which ultimately contains more details) could already be incorporated into the second paragraph.

We restructured Section 3 in the revised manuscript to make it more concise.

Lines 225-226: Could you present any quantitative numbers on this minor influence?

The number nebulizing efficiency and the relative humidity are not changing (within a few percent) when varying the cooled section temperature in the low positive temperature range (i.e. 0 to 5°C). However, the efficiency increases and the r.h. decreases for sub-zero temperatures. Operating the nebulizer at a sub-zero cooled section temperature resulted in regular failures of the nebulizer, most likely due to water freezing in the cooled section.
We changed added "a few percent" to the sentence.

Section 4

Lines 263-266: What about adding the size distribution of the fullerene in a figure, to show that the difference in fullerene MAC's could be explained by differences in size?

We added a figure of the size distributions to the manuscript.

Section 5

The results of the snow sample measurements could be highlighted even more. You have 33 samples from one snow season. How do they differ from one another? In Fig. 7 you show how the BC snow concentration varies over this season, what about the air concentration of BC from the station? (it is mentioned that no correlation was found between the snow and air BC concentrations. I would argue that this is shown, possibly in fig. 7 or a different figure). Is there any seasonality in the size distribution of BC particles from snow? I'm not convinced that there is such a clear decrease in the size distribution of the larger BC particles, please elaborate on this to further convince the reader that this is the case. (It is true that since you have taken relatively fresh snow samples, there would be no thawing and freezing cycles, possibly leading to larger BC particles). One example comes to mind where fresh snow BC size distributions were measured (Sinha et al., https://doi.org/10.1002/2017JD028027). Although in a different setting, please put your results into context by comparing with this study.

We agree with the reviewer that the presentation and discussion of the snow sample results in the context of concurrent meteorological and ambient aerosol measurements is too short or even lacking in the manuscript. That's why we have replaced Fig. 7 by a stacked plot (Fig. 8 in the revised manuscript) showing maximum and minimum temperature, sunshine duration, precipitation, snow height as well as the air BC mass concentration records over the time period the snow samples were taken.

To discuss the measured rBC snow concentrations in context with these other measurements we added the following paragraph to Section 5:

"This BC concentration is shown in **Error! Reference source not found.**c in conjunction with the eBC mass concentration of ambient air that is routinely measured by the German Federal Environment Agency using a Multi-Angle Absorption Photometer (MAAP). Also, a selection of meteorological data is presented in **Error! Reference source not found.** to highlight the trends in ambient temperature, sunshine duration, snow precipitation and snow height over the period the snow samples were collected at the UFS station. Although there is no clear correlation between the fresh snow samples and ambient air eBC mass concentration, the enhanced air eBC mass concentration observed end of March and beginning of April might have resulted in additional deposition of BC particles in the snow surface that is reflected - with a time lag of several days - in the measured snow refractory BC mass mixing ratio. Interestingly, this period of higher air eBC concentration is distinguished by a low precipitation activity, long sunshine periods as well as frequent daily maximum temperatures above the melting point that resulted in frequent thaw/freeze cycles and, consequently, a gradual decrease of the snow height by 30 to 40 cm. All in all, the enhanced air eBC concentration in conjunction with the meteorological conditions would favor enhanced BC mass concentrations in the fresh snow samples collected after precipitation events within this period or shortly after."

Actually, from the 33 samples we use only 31 samples in the revised manuscript as two samples were discarded due to inexplicable high rBC mass concentrations and absorption coefficients (factor 5 to 10 outside the 95th percentile of the other samples). To investigate a possible seasonality in the measured rBC size distributions of the snow samples, we divided the 31 samples into the three periods (1) November to January with 11 samples, (2) February and March with 11 samples, and (3) April and May with 9 samples. For each period the average rBC size distribution is calculated and is plotted in the updated Fig. 8 (which is Fig. 9 in the revised manuscript).

We thank the reviewer for noticing the Sinha et al. paper. In Fig. 8 of their paper the BC mass size distribution of a fresh snow sample is shown that was collected after a snowfall event at Ny-Ålesund, Svalbard, Norway. We also found the study by Schwarz et al. (2013) very useful as this study presents an averaged rBC mass size distribution of snow samples collected shortly after snowfall events in Colorado, USA (Fig. 1 in their study). Our averaged fresh snow sample size distributions peak at similar mass median diameters (MMD) between 194 and 227 nm compared to the 223 ± 28 nm of the Sinha et al. study and the ~220 nm of the Schwarz et al. study. In addition, our BC mass size distributions indicate a non-lognormal shoulder at the upper size limit of our measurement that is in a very good agreement with the Schwarz et al. (2013) samples where the BC mass size distribution was measured up to 2 μm (see Fig. 9 in the revised manuscript which shows a comparison with the Schwarz et al. data). As it is discussed by Schwarz et al. (2013) such a size distribution reflects the typical atmospheric BC distribution at remote locations that is altered by agglomeration and size selection processes during snow formation in the atmosphere.

We also divided the snow samples into the three periods Nov-Jan, Feb-Mar, and Apr-May and present the averaged size distributions of these three periods in Fig. 9 of the revised paper. This clearly shows that here is no significant seasonality in the snow BC data.

We have added the following paragraph to Section 5 to include these comparisons and their

conclusions:

"Figure 9 shows corresponding mass size distributions of the refractory BC concentrations shown in Figure 8c averaged over the periods November to January, February and March, as well as April and May. For comparison purposes, the average size distributions are normalized by the corresponding total mass concentration Mtotal, which was deduced from a lognormal fit. The SP2-derived refractory BC mass size distribution only includes particles up to a mass-equivalent diameter of 560 nm, which means that larger BC particles are not recorded by the SP2. However, the average BC mass size distributions have distinct mode maxima at the mass median diameters (MMD) of 227, 194, and 222 nm for the Nov-Jan, Feb-Mar, and Apr-May periods, respectively. This indicates no strong seasonality in the snow BC mass size distribution even in the Apr-May period where the BC mass concentration in the snow was significantly enhanced (Figure 8c). This further implies that indeed fresh snow was sampled which hasn't experienced thaw/freeze cycles severe enough to induce an agglomeration of the BC particles in the top snow layer. This conclusion is further supported by comparing the average BC mass size distributions of our snow samples with the BC mass size distribution of a fresh snow sample collected after a long-lasting snowfall event at Ny-Ålesund, Svalbard, Norway by Sinha et al. (2018) and with the averaged BC size distribution from five snow samples collected after three snowfall events in the semi-rural and rural surroundings of Denver, CO, USA by Schwarz et al. (2013). Our average fresh snow sample size distributions peak at similar MMD between 194 and 227 nm compared to the 223 ± 28 nm of the Sinha et al. study and the ~220 nm of the Schwarz et al. study. In addition, our size distributions indicate a non-lognormal shoulder at the upper size limit of the SP2 measurement that is in a very good agreement with the Schwarz et al. (2013) samples where the refractory BC mass size distributions were measured by a SP2 with modified detector gains up to 2 µm (see Figure 9). As pointed out by Schwarz et al. (2013) such snow BC mass size distributions reflect the typical atmospheric BC mass size distribution that is observed at remote locations altered by agglomeration and size selection processes during snow formation in the atmosphere. The good agreement between the mass size distributions of our snow samples and the average distribution of the Schwarz et al. (2013) samples allows us to estimate the refractory BC mass that is contained in the large particle size shoulder outside our measurement range. According to Schwarz et al. (2013) a fraction of 28% of the total BC mass can be attributed to particles with mass-equivalent diameters larger than 600 nm. A mass correction factor of 1.39 is therefore applied to the SP2-derived refractory BC snow concentrations in the following analysis."

What about the dust concentrations? How did the described Saharan dust episodes influence the BC mass (and the snow samples)?

Although we cannot directly measure the dust concentrations in the snow samples, we can draw some conclusion by analyzing the spectral signature of the snow mass specific absorption cross section of the non-BC particles, i.e. after subtraction the spectral cross section that is expected from the SP2 mass data using the MAC of Fullerene soot (Equation 3 of the revised manuscript). We added a figure (Fig. 12 in the revised manuscript) that shows a comparison of the non-BC spectral absorption cross section with laboratory absorption data for Saharan dust particles. Thus, Saharan dust is a good candidate to explain the observed non-BC light absorption in the snow samples. We did the same comparison with laboratory data of OC, which reveals that OC could also explain the observed non-BC absorption - not only in terms of wavelength dependence but also in terms of variability.
The whole paragraph on the analysis of the spectral absorption analysis has been rewritten to make our argumentation of a significant influence of non-BC particles on the light absorption in snow clearer.

Through the measurement set-up that the authors present, they should have the data necessary to directly derive MAC values for the particles in the snow. How come this was not done? I believe one of the other referees also commented on this. Either the authors provide these MAC values also, and compare (and discuss) that in the manuscript with the data that they already have. If this

is not possible, then it should be stated why, and more emphasized why the approach used currently in the manuscript is utilized. There is evidently other impurities other than the BC particles, which will influence the MAC value for the particles in the snow, but with the other instrumentation available, it should assist in describing those particles (i.e. the OC content).

We did a thorough reanalysis of the snow samples following this strategy:

1. The snow mass specific absorption cross section $\sigma_{abs}$ [m²/mL] was deduced from the absorption coefficient $b_{abs}$ [m⁻¹] using the nebulizer flow settings and the nebulizing efficiency (Eq. 1 of the revised manuscript):

$$\sigma_{abs} = 10^{-3} \cdot b_{abs} \cdot R_{neb}/R_{pp} \cdot \varepsilon_{neb}^{-1}$$

2. This snow mass specific absorption cross section is plotted as a function of the refractory BC mass concentrations in a new figure (Fig. 10 of the revised manuscript). Linear regression fits to the data then give the BC mass specific MAC of the snow samples. We found MAC values that are up to a factor of two larger than the MAC of Fullerene soot. A table is added to the manuscript contrasting the mass and optical properties (including the MAC) of Fullerene soot with those of the snow samples.

3. To be comparable with other studies (e.g. Doherty et al., 2010), we calculated the equivalent BC mass concentration $c_{BC}^{equiv}$, i.e. the amount of BC that would need to be present in the snow to account for the measured absorption, from $\sigma_{abs}$ using the MAC of Fullerene soot (Equation 2 of the revised manuscript):

$$c_{BC}^{equiv} = \sigma_{abs}/\mathrm{MAC}_{FS}$$

4. We plotted $c_{BC}^{equiv}$ as a function of the refractory BC mass concentration in a new figure (Fig. 11 of the revised manuscript) and found a good correlation of both concentrations but with correlation coefficients of 2.0, 1.9, and 1.4 for 405, 532, and 658 nm, respectively (note that $c_{BC}^{equiv}$ is a function of the wavelength). We conclude from this that there is additional non-BC light absorbing mass in the snow, which is correlated with the BC mass and which has a strong wavelength dependence between the green and the red part of the visible spectrum. This already indicates mineral dust and organic (brown) carbon as possible carriers of this additional absorption.

5. We calculated the snow mass specific absorption cross section of the non-BC particles, $\sigma_{abs}^{nonBC}$ (Equation 3 in the revised manuscript):

$$\sigma_{abs}^{nonBC} = \sigma_{abs} - c_{BC}^{SP2} \cdot \mathrm{MAC}_{FS} \cdot 10^{-9}$$

We added a figure (Fig. 12 in the revised manuscript) that shows the statistical analysis of the $\sigma_{abs}^{nonBC}$ for the snow samples and that compares this non-BC spectral behavior with laboratory data for Saharan dust and organic (brown) carbon (see answer to the previous comment).

Line 313: What evidence is there to show that this one sample from March 10th is representative for all of the snow samples? (This sample contains a low amount of BC compared to the other samples and has an enhancement factor of 2.7 compared to the 2.34 presented for the others samples). Either present some evidence that this is representative for all of the samples or emphasize in the manuscript that these additional analyzes (presented in the following paragraphs) were only done to the one sample, and so it is difficult to draw conclusions for all of the snow samples. Ideally, I would argue that additional snow samples would be analyzed from different times during the season in the same way as the one sample from March 10th.

Actually, after the reanalysis of the samples, which includes the correction for missing BC mass in the SP2 measurement, the March 10 sample has a mass concentration $c_{BC}^{SP2}$=2.8 ng/mL and an equivalent BC mass concentration of $c_{BC}^{equiv}$=6.0 ng/ml for λ=532 nm, which gives an enhancement factor of $\gamma = 2.1$. It is therefore representative for $\gamma$, but is on the lower side concerning the $c_{BC}^{SP2}$ and $c_{BC}^{equiv}$ concentrations. We added a mark in the $c_{BC}^{equiv}$ versus $c_{BC}^{SP2}$ plot (Fig. 11 of the revised manuscript) to indicate the representativeness of this sample. As mentioned above, further analyses (especially with the ESEM) are very time consuming and expensive, which we couldn't be invested within the scope of this pilot study. This is now clearly stated in the revised manuscript:

"While these results give a detailed look into the physical and chemical nature of the of the particles that might contribute the light absorption in the March 10 snow sample, they cannot used to draw conclusions for all snow samples. Here, further analyses are required that couldn't conducted within the scope of this pilot study."

Lines 320-326: Although you discuss it further in the following paragraph, please include some sentence (or sentences) of what these specific results indicate about the snow samples?

Will be added to the revised manuscript.

Line 329: It would be interesting to have some information (even if it is hypothetical) on where this biological information originate from? E.g. Local or long distance?

We changed the terminology in the discussion of the ESEM and WIBS analysis by substituting the term "biological" with "biogenic". Although we have indications of biological components in the ESEM analysis, like bacteria, pollen and spores, we cannot conclude that all biogenic EDX patterns are due to microorganisms or their fragments. We added a paragraph on possible origins of the biogenic material found in the March 10 sample, mainly in the context of the observed non-BC light absorption by brown carbon.

"However, one question that arises from the above findings is whether the biogenic particles found in the March 10 snow sample can be attributed to BrC, which was shown to be a good candidate for explaining the additional light absorption in the snow samples (**Error! Reference source not found.**). The term "brown carbon" is not clearly defined or characterized and is mainly related to a strong wavelength dependence of the visible light absorption observed in these materials. From a chemical perspective, BrC can generally be divided into humic-like substances (HULIS) and tar balls (Wu et al., 2016). HULIS can be characterised mainly as a mixture of macromolecular organic compounds with various functional groups and are expected e.g. in oxidation processes of biogenic precursors (Wu et al., 2016). Tar balls are emitted from biomass burning and are of spherical, amorphous structure and are typically not aggregated. Moreover, light absorbing organic material and HULIS can be formed from the water-soluble fraction of biomass burning aerosol compounds, and is therefore suggested as an atmospheric process for the formation of light absorbing BrC in cloud droplets (Hoffer et al., 2004). Further examination of snow samples from different locations as well as systematic investigations on the optical behaviour of biogenic particulate matter is therefore necessary to evaluate the influence of biogenic (including biological), BrC and mineral dust on the aerosol absorption properties in the visible spectral range."

All the comments given in the technical corrections are addressed in the revised manuscript.

**Technical corrections:**

Lines 29-30: This opening sentence is not structured well. Please revise for a better read.

Line 32: Please remove 'packs' from ice packs. Could say snowpack, but not ice packs here.

Line 32-33: You could argue that a better reference here would be Warren and Wiscombe 1980 https://doi.org/10.1175/1520-0469(1980)037<2734:AMFTSA>2.0.CO;2, look into details of Doherty et al. 2010.

Lines 33-34: How is this sentence different than the previous sentence? I would think that it is better to have this sentence earlier.

Lines 35-36: How does 'this reduction' contribute to the snow-albedo feedback? Please include in the manuscript. What metamorphosis?

Line 41: What 'BC amounts' are you referring to? Please specify.

Lines 44-46: How are permafrost regions also affected? Unclear what you mean how they are affected? Also, after reading the rest of this paragraph, I would argue that you should remove this sentence. Since the rest of the paragraph discusses the Arctic, and these other 'areas' are not brought up again until later in the introduction, it could come then instead.

Line 50: As far as I remember Flanner et al. (2007) did not present any measurements, but based their modeling work on measurements instead.

Line 53: I would argue that you either introduce what the term 'soot' refers to, or stick with only discussing BC.

Line 56: Ice sheet instead of 'ice shield.' Please change also 'extend' to extent.

Lines 55-60: Concerning the Greenland ice sheet, there are also new papers on this topic of impurities, which could be added here (e.g. https://doi.org/10.5194/tc-10-477-2016; https://doi.org/10.5194/tc-11- 2491-2017).
Line 60: Please capitalize a in 'arctic.'
Line 61: Doherty et al. (2010) is already referenced to in the beginning of the sentence.

Line 68: How high amounts of dust? Would be more informative to actually reference to some numbers on this.
Line 69: To my knowledge, Bolch et al. (2012), is incorrectly referenced to here. No studies of BC nor dust were conducted in that paper.
Line 73: Remove 'the' before light absorbing particles.

Lines 77-80: I find this sentence confusing, please rewrite. Mixing the optical method (Doherty et al., 2010), and then the thermos-optical analysis, with the previous sentence about MAC causes some of the confusion. The following paragraph (lines 81-93) dig deeper in each analysis technique and that is appropriate, but the order of this seems strange, in light of the previous paragraph. Actually, I think you could delete the sentences in lines 75-80, and jump right into line 81 and an explanation of the methods (after current sentence ending on line 75).

Lines 92-93: Either remove this sentence or add more information on other instruments and

protocols out there (e.g. DRI and Improve-protocol). I would vote for removing this sentence, I do not think it is very crucial information.

Line 97: I would argue that you do not need quotation marks around Fullerene.

Lines 98-100: This sentence is basically a repeat of the first sentence of this paragraph, please remove.

Lines 102-103: I generally agree with this statement that not much have been reported on the light absorbing properties. But, there has been some publications on this topic addressing it directly and indirectly, e.g. Schwarz et al., 2013 https://doi.org/10.1038/srep01356; Zhang et al., 2017 http://dx.doi.org/10.1016/j.scitotenv.2017.07.100; Dal Farra et al., 2018 doi: 10.1017/jog.2018.29; Dong et al., https://doi.org/10.5194/tc-12-3877-2018. Please add and discuss these references.

Line 104: Please remove 'solar' before albedo.

Lines 103-106: Please clarify the structure by checking the structure. As it currently stands, it is not clear what the main point of the sentence is.

Lines 106-108: Please change this sentence according to the forthcoming changes made for lines 102- 103.

Lines 109-110: The second half of this sentence (starting after 'but) I find problematic. Similar to the comment in lines 102-103, I do believe this topic has been addressed in the literature. For example: Kaspari et al., 2014 (that you already referenced to earlier in the manuscript); Skiles and Painter, 2016 doi: 10.1017/jog.2016.125; Schmale et al., 2017 DOI: 10.1038/srep40501; Zhang et al., 2018 https://doi.org/10.5194/tc-12-413-2018, 2018. Please adjust your claim by including these references on this topic.

Line 137: The fact that the snow samples were collected at 'platform 7' does not add any information to a reader unfamiliar with UFS. Please either elaborate on this, or remove.
Line 147: What does 'Enhanced' refer to? Please explain.

Line 150: I do not find the flow rate for this peristaltic pump anywhere. Please add it. Line 184: Please remove the double reference to 'Fischer and Smith (2018).'

Lines 196-198: Do you mean that the solution was prepared in the same way as in Schwarz et al? If so, please correct. Also, I believe the reference should be Schwarz et al. (2012) and not (2010) as it currently reads.

Line 204: How did you 'drop' 30 mL of fullerene solution onto the quartz filters? Please explain more.

Line 260: What did Zhou et al. (2017) refer to with MACreal? Please clarify.

Line 276: Remove 'before' at the end of the sentence.

Line 294: The presented enhancement factors for the different wavelength appear to be averages, please clarify this.

Lines 315-319: The instruments and methodology presented here should be described in section 2. Lines 344-345: This information should be moved to section 2.

Figure 2. I'm not sure how needed this figure is. I actually think that this figure could be integrated into fig. 1.

Figure 8. Why is there is a data gap around 280-300 nm?

Figure 9. This figure is quite busy right now. Could the data points be zoomed in on more? And could the data points be made smaller?

---

## Author Response (AR2)

We thank the Co-Editor for his helpful comments.

In the following we give answers to his comments in blue.

The paper needs an additional and thorough editorial read. Especially the language needs some improvement (as also noted by Reviewer #2) and I would recommend a proof-reading by a native speaker. I have had no time to correct all language issues.

The manuscript was native speaker proof-reading.

Please study once more the manuscript preparation guidelines (https://www.atmospheric-chemistry-and-physics.net/for_authors/manuscript_preparation.html) and follow the rules especially with regard to units (e.g. units should be formatted with negative exponents) and abbreviations within the text.

Done.

Line 33: Earth should be capitalized.
Done.
Line 51: 'ration' -> 'ratio'
Done.
Line 54: Add 'on' before 'optical properties'.
Done.
Line 55: I would replace 'from giving' by 'to give'.
Done.
Line 61: I would put the "integrating-sandwich with integrating sphere technique' in italics and/or mark how the abbreviation ISSW results from this (it is not clear to me).
Done.
Line 64: I guess this relates to ISSW, so maybe replace 'their method' by 'the ISSW method'.
Done.
Line 66: A small Latex hint for here and throughout the manuscript: Put '\rm' into the sub- or superscript when you like to add text or abbreviations into variables, symbols or equations (so e.g. $\sigma_{\rm abs}$).
Done.
Line 72: Please add that the Schneefernerhaus is located in Germany or in the German Alps.
Done.
Line 76: Here and throughout the manuscript: fullerene should not be capitalized (https://en.wikipedia.org/wiki/Fullerene).
Done.
Line 79: 'Sect. 3' -> 'Section 3' (it is the beginning of a sentence)
Done.
Line 82: Add 'in' before Sect. 5.
Done.
Line 98: The ambient aerosol data measured by the OPC and CPC seemed to be used later on. Suggest to remove this. Please mention the model and manufacturer of the MAAP (as has been done for all the other instruments later on).
Done.
Line 107: KIT is not defined yet.
Defined now.
Line 133: The exhaust line is not shown in Fig. 1.
Is now shown in Fig. 1
Line 168: Reference for Nanopure water is missing.

Replaced "Nanopure" by "ultrapure", because this is the more generic term.

Line 194: The CPC type and manufacturer is not mentioned.

Mentioned now.

Line 209: This sentence is unclear (a verb is somehow missing).

Rephrased.

Line 271: Here and throughout the manuscript: Please ensure that you figures are discussed in order of appearance.

Checked.

Line 281: Maybe mention that these numbers are 'mean +/- std' (or reference to Table 1 once more).

Inserted a reference.

Line 308: The referencing to the German Federal Environment Agency is repetitive (see method part).

Used the abbreviation here.

Line 387: If I understood it correctly, gamma is not a correlation coefficient but rather a slope (derived from a linear regression). Also later, gamma is suddenly called enhancement factor. I would suggest to properly introduce it here.

Clarified.

Page 12, first part of the second paragraph: This is part of the methods and I would suggest move it to the front (maybe as an own subsection).

Moved to the experimental section.

Line 453: 'Figure 14 Figure 16' -> 'Fig. 14 to Fig. 16.'

Fixed.

First paragraph on page 13: How did you derive the fraction of biogenic particles or what definition did you use to discriminate biogenic particles using the WIBS4? It would be nice to have a dedicated figure for this result (maybe for the supplement?).

Added a figure to the supplementary material (S6).

- The statement on the data availability is missing (https://www.atmospheric-chemistry-and-physics.net/about/data_policy.html).

Added.

- Figure 4: The figure is difficult to read when having a black-and-white printout.

Changed.

- Figure 12: The right axis is not described in the figure caption.

Changed.

- Figure 13: For me it looks like that some of the green arrows have shifted and are not really pointing towards the green dots.

Fixed.

- Please add in the respective figures what kind of linear regressions you have used. For some figures it also looks like that the regression lines were forced through zero. If so, this should be mentioned.

Added.

- Your manuscript has quite large number of figures (as also pointed out by reviewer #3). You could consider to move some of the more technical figures to the supplement (e.g. Fig. 3, 4, 5, 8).

Moved part of the figures to the supplementary material.

We thank the reviewer for the helpful comments.

In the following we give answers in blue.

Referee Review of manuscript

I compliment the author's on the revision of their manuscript. It is more clearly written and much (much) easier to read. Now the findings from this work can be readily appreciated with out distractions. The authors have addressed the concerns that I raised in my original review and thus I recommend publication. I do have a couple of small editorial comments the authors might consider.

Side note: The authors make convincing argument regarding the variability of MACs. With such variability, one has to wonder if we'll ever be able to better quantify BC forcing beyond where we are today. This is, however, a subject for a another paper.

17 figures is a lot of figures for the main manuscript. There are several figures that could be located in a supplemental. For example, Figures 3, 4, 5, 6, as these refer to performance metrics. Additionally, perhaps only an example of a SEM image (along with the EDX-spectrum) is needed for the main manuscript with the remainder, again, being located in the supplement.

Reorganized the figures and moved part of the figures to the supplementary material.

Page 2, line 51: please correct "ration" to "ratio"

Done.

page 2, line 77: please remove "also" along with the comma.

Done.

Page 7, line 48: Please be explicit. Who is "They"? Is it Zangmeister or Schwartz?

Corrected.

Page 8, line 293: Please consider presenting absorption coefficient units as Mm^-1 (superscript "-1) versus 1/Mm.

Done.

Page 9, Lines 327 - 329: Can you provide an example (e.g., expected mode size) of agglomeration of BC particles that have experienced thaw/freeze cycles? How much of a shift is expected? From a dM/dLogD point of view, this could be significant.

We did not identified samples that had experienced strong thaw/freeze cycles.

Page 10, Ling 369 - 372: Can the SP2 help here in further quantifying the contribution of rBC mixing state [coating thickness (in the limit of a core-shell assumption)] to the observed MACs? As the authors point out, the lensing-induced enhancement of light absorption is strongly dependent upon coating thickness. It seems to this reviewer, that a quick examination

of the BC mixing state would augment the discussion on the influence of non-BC aerosol particles.

In principle this is possible, but we operated the SP2 without the scattering channel to keep the data volume low. The re-aerosolized snow samples show a high concentration of light scattering particles.

**Compare Results**

| Old File: | | New File: |
|---|---|---|
| **acp-2018-1307-manuscript-version3.pdf** | versus | **acpd-2018-1307_review_08022019.pdf** |
| **28 pages (3.78 MB)** | | **26 pages (3.41 MB)** |
| 16.06.19, 09:38:10 | | 02.08.19, 17:34:12 |

**Total Changes**

**531**

Text only comparison

**Content**

323 Replacements

105 Insertions

103 Deletions

**Styling and Annotations**

0 Styling

0 Annotations

Go to First Change (page 1)

[revised manuscript text omitted]